# The synaptonemal complex has liquid crystalline properties and spatially regulates meiotic recombination factors

Ofer Rog[1,2,3*†], Simone Köhler[1,2], Abby F Dernburg[1,2,4,5*]

[1]Department of Molecular and Cell Biology, University of California, Berkeley, Berkeley, United States; [2]Howard Hughes Medical Institute, Chevy Chase, United States; [3]Department of Biology, University of Utah, Salt Lake City, USA; [4]Biological Systems and Engineering Division, Lawrence Berkeley National Laboratory, Berkeley, United States; [5]California Institute for Quantitative Biosciences, Berkeley, United States

*For correspondence: ofer.rog@utah.edu (OR); afdernburg@lbl.gov (AFD)

Present address: †Biology Department, University of Utah, Salt Lake City, United States

Competing interests: The authors declare that no competing interests exist.

**Abstract** The synaptonemal complex (SC) is a polymer that spans ~100 nm between paired homologous chromosomes during meiosis. Its striated, periodic appearance in electron micrographs led to the idea that transverse filaments within this structure 'crosslink' the axes of homologous chromosomes, stabilizing their pairing. SC proteins can also form polycomplexes, three-dimensional lattices that recapitulate the periodic structure of SCs but do not associate with chromosomes. Here we provide evidence that SCs and polycomplexes contain mobile subunits and that their assembly is promoted by weak hydrophobic interactions, indicative of a liquid crystalline phase. We further show that in the absence of recombination intermediates, polycomplexes recapitulate the dynamic localization of pro-crossover factors during meiotic progression, revealing how the SC might act as a conduit to regulate chromosome-wide crossover distribution. Properties unique to liquid crystals likely enable long-range signal transduction along meiotic chromosomes and underlie the rapid evolution of SC proteins.

## Introduction

In most eukaryotes, chromosome pairing during meiosis culminates with synapsis, defined as the assembly of synaptonemal complexes (SCs) between homologous chromosomes (homologs). Upon meiotic entry, prior to pairing and synapsis, replicated chromosomes reorganize around a central 'axis,' a linear structure comprised of cohesin complexes and associated meiosis-specific proteins. Once chromosomes establish local interactions with their homologs through recombination or other mechanisms, the central region of the SC nucleates and assembles progressively between paired axes, resulting in close side-by-side alignment of homologous chromosomes along their entire lengths (*MacQueen et al., 2002*; *Page and Hawley, 2004*; *Rog and Dernburg, 2015*). SC assembly is required for stable interhomolog pairing, normal levels of crossover (CO) recombination, cell cycle progression, and faithful chromosome segregation (*Page and Hawley, 2004*).

While disruption of the SC dramatically alters the CO distribution in a variety of organisms, the functional contribution of the SC to CO regulation remains hotly debated. Evidence from budding yeast has suggested that the SC may be a passive 'glue' that merely stabilizes the physical pairing of homologs to permit efficient CO formation (*Börner et al., 2004*; *Fung et al., 2004*; *Zickler and Kleckner, 2015*). However, evidence from *C. elegans* has indicated that the SC plays a direct role in the conserved phenomenon of *crossover interference*, which distributes COs in a non-random, widely spaced pattern along each chromosome (*Hayashi et al., 2010*; *Libuda et al., 2013*).

**eLife digest** The genetic information in cells is encoded within long molecules of DNA called chromosomes. In most human cells, the two copies of each chromosome – the one inherited from our mother and the one from our father – are physically separated and behave independently. However, in the reproductive cells that give rise to eggs or sperm, each chromosome must pair with its partner. Pairing first occurs at one or more positions along each chromosome. This triggers a protein-based polymer called the "synaptonemal complex" to assemble between the paired chromosomes, and then spread along the interface between the partners until they are fully lined up side-by-side. Chromosomes in reproductive cells must pair in this particular way to exchange genetic information and generate new combinations of traits.

The synaptonemal complex was first observed over 60 years ago, but it remains enigmatic. Though its structure is highly ordered and looks very similar in different organisms from yeast to humans, little is known about how this polymer forms or what it does between chromosomes. Some evidence has suggested that the synaptonemal complex helps to regulate how much information can be transferred between each pair of chromosomes, but not all studies have supported this conclusion.

Several lines of evidence suggest that the synaptonemal complex might be fundamentally different from other protein-based polymers, such as those that form filamentous skeletal structures within cells, namely actin filaments and microtubules. Now, Rog et al. have tested the idea that the synaptonemal complex might actually have liquid-like properties, despite its highly ordered appearance.

The experiments showed that the proteins that make up the synaptonemal complex in yeast, worms and fruit flies are weakly bound to each other and can move around within the assembled structure. These are considered to be defining properties that distinguish liquids from solid materials. Together with its regular, repetitive organization, these findings indicate that the synaptonemal complex behaves like a liquid crystal. This intriguing class of materials has properties between those of conventional liquids and those of solid crystals, and is particularly sensitive to environmental conditions.

Rog et al. believe that this discovery helps to explain how signals are transmitted along the length of chromosomes to regulate the transfer of genetic information. In support of this idea, further experiments showed that proteins that are required for this recombination process were also found within the synaptonemal complex. As reproductive cells transition from one stage of their development to the next, these proteins abruptly move to a new location, indicating that a switch-like signal rapidly spreads throughout the synaptonemal complex.

Together the findings suggest that the liquid crystal-like properties of the synaptonemal complex allow signals to be transmitted along the interface between pairs of chromosomes. The next challenges are to understand what triggers these signals and to explore whether they are based upon physical or chemical changes within the synaptonemal complex. Further research is also needed to uncover how this information is propagated along the length of a chromosome.

The SC was initially observed sixty years ago by thin-section transmission electron microscopy (TEM), which revealed electron-dense linear structures at the interface between meiotic chromosomes (*Moses, 1956*). Subsequent observations of meiocytes from diverse eukaryotes led to a consensus view that the SC is a symmetrical, tripartite structure, with two parallel, darkly-staining lateral bands flanking an electron-lucent, transversely striated, central region (*Westergaard and von Wettstein, 1972*and *Figure 1B*). Electron microscopy, superresolution fluorescence imaging, and protein-protein interaction analysis have clearly indicated that the proteins that make up this complex form a highly ordered, periodic structure with bilateral symmetry (*Schild-Prüfert et al., 2011*; *Schücker et al., 2015*). Serial-section TEM analysis also led to the discovery of 'recombination nodules' associated with SCs at sites of genetic exchange (*Carpenter, 1975*), and specific recombination factors have been localized to these sites by a variety of cytological methods. However, how the SC

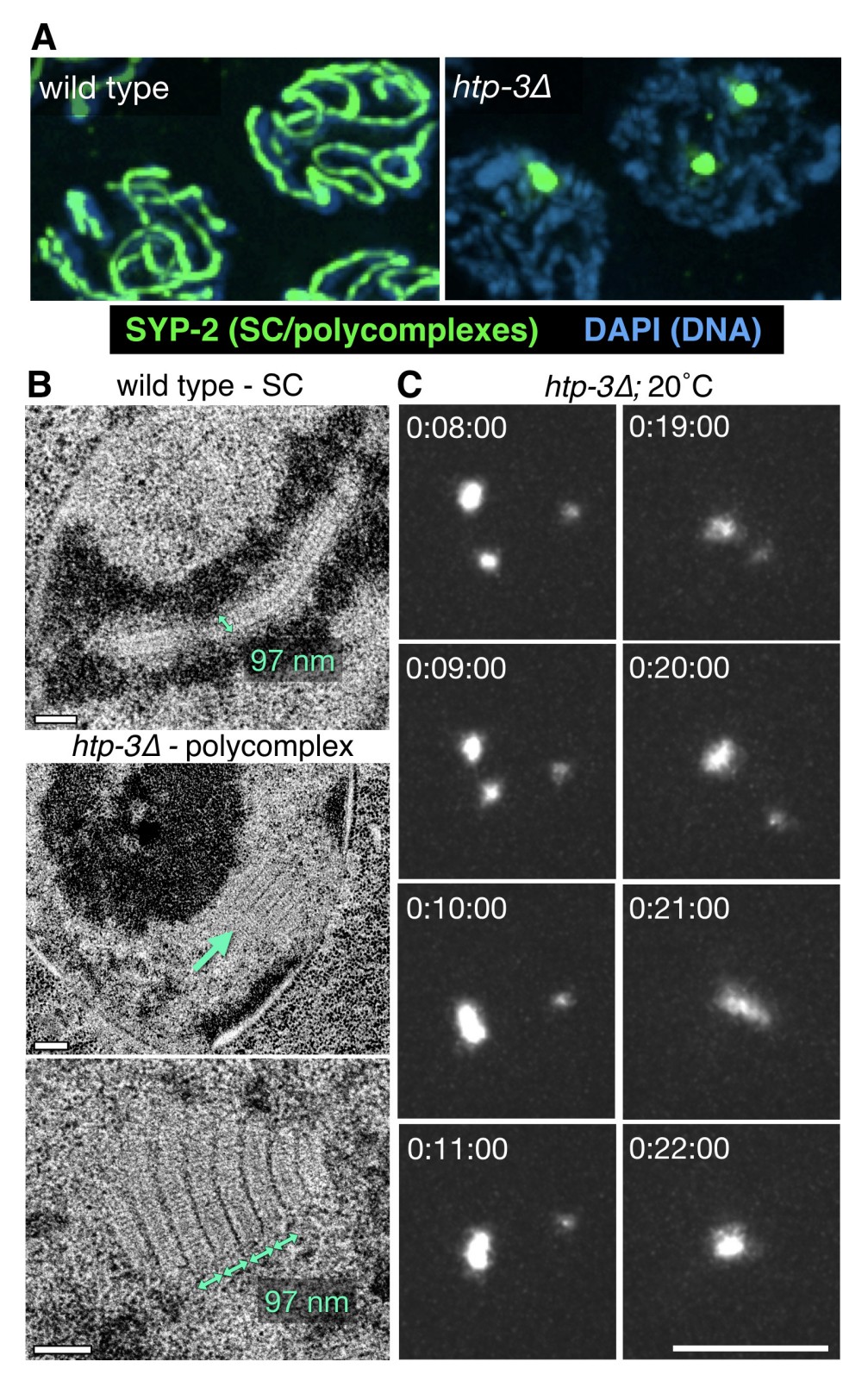

**Figure 1.** Polycomplexes are 3D lattices of SC proteins that exhibit liquid-like behaviors. (**A**) Fluorescence micrograph showing mid-prophase oocyte nuclei from wild type and *htp-3(tm3655)* hermaphrodites, immunostained for SYP-2. Whereas in wild type animals SCs load between homologous chromosomes and appear as long filaments, in the absence of HTP-3 SC proteins form one or more large bodies that contain all of the known SC central region proteins. See *Figure 1—figure supplement 1A* for images of the entire gonads. (**B**) Representative transmission electron micrographs. *Figure 1 continued on next page*

*Figure 1 continued*

The top image shows a single nucleus from a wild type hermaphrodites, with a synapsed chromosome pair (darkly-staining material flanking the SC is chromatin). The middle image shows a single nucleus from a *htp-3(tm3655)* hermaphrodite, with a polycomplex indicated by the green arrow. The darkly staining region in the center of the nucleus is the nucleolus. The bottom image shows a higher-magnification view of a polycomplex from a different nucleus. The distance between parallel darkly-staining bands is 97 nm, identical to the width of SCs that normally form between homologous chromosomes. Notably, these polycomplexes do not contain any of the known chromosome axis components, including cohesins and the HORMA domain proteins HTP-3, HIM-3, HTP-1 and HTP-2 (*Severson et al., 2009* and our observations). This implies that the electron-dark lateral bands of the SC do not correspond to the chromosome axis, as has been long presumed, but are instead part of the structure formed by the central region proteins. See *Figure 1—figure supplement 2A* for serial sections of the polycomplex in the lower panel. Scale bars = 0.2 μm. (C) Projection images showing a single nucleus from a live recording of a *htp-3(tm3655); GFP-SYP-3* hermaphrodite, at selected time points. Elapsed times are indicated as hours: minutes:seconds. Polycomplexes continually undergo deformations and fusions. Here, fusion of two initially separate polycomplexes is observed between 0:09:00 and 0:10:00, and again between 0:20:00 and 0:21:00. The full recording is shown in *Video 1*. Scale bar = 5 μm.

The following figure supplements are available for figure 1:

**Figure supplement 1.** Further characterization of polycomplexes and heat-induced SC aggregates.

**Figure supplement 2.** Further characterization of polycomplexes.

assembles between chromosomes, and how this polymer might govern or respond to meiotic recombination are largely mysterious.

Synapsis of homologous chromosomes typically requires both nucleation factors and structural proteins that localize throughout the SC, which spreads along the interface between aligned chromosomes (*Page and Hawley, 2004*; *Zickler and Kleckner, 2015*). Structural components, defined here as proteins required for SC assembly, have been identified through genetic screens and/or immunocytochemistry in a variety of organisms. Three to five such proteins have been characterized in budding yeast, *Drosophila*, *C. elegans*, and mammals, and at least one SC protein has been identified in plants. However, it is not yet clear whether the full cohort of SC structural elements has been defined in any organism. While the dimensions and appearance of the SC are conserved among diverse phyla, these structural proteins show remarkable divergence in their lengths and primary sequences. All known SC structural proteins contain extensive regions that score highly on coiled-coiled prediction scales. Together with the transverse striations observed by TEM, this has led to a prevalent view of the SC as a stable, 'crosslinking' polymer assembled through coiled-coil interactions, perhaps resembling intermediate filaments. However, other observations have hinted at a more dynamic structure. In particular, evidence from budding yeast has revealed that SCs can incorporate newly translated proteins after assembly (*Voelkel-Meiman et al., 2012*), consistent with dynamic exchange of subunits. Additionally, in many organisms SCs are known to undergo 'synaptic adjustment,' post-assembly progressive reorganization that tends to simplify their topology (*e.g.*, the extent of loop structures and unsynapsed axes is minimized) (*Page and Hawley, 2004*; *MacQueen et al., 2005*; *Henzel et al., 2011*) – a behavior that implies extensive rearrangement of subunits after initial assembly of the complex.

Here we probe the biophysical properties of the SC in vivo, and show that it exhibits liquid-like behaviors and likely assembles through a regulated coacervation process. We further report that SCs in *C. elegans*, budding yeast, and *Drosophila* are rapidly and reversibly dissolved by aliphatic alcohols, indicating that their integrity relies on weak hydrophobic interactions. Finally, we demonstrate that the dynamic localization and interdependence of two essential CO factors is recapitulated by chromosome-free assemblies of SC proteins, demonstrating how an SC compartment may dynamically partition enzymatic activities to regulate meiotic recombination.

## Results

### SCs exhibit liquid-like behaviors

The SC normally assembles between the paired axes of meiotic chromosomes (*Couteau et al., 2004*; *Severson et al., 2009*; *Kim et al., 2014*). Note that throughout this work we use terminology that differentiates between the SC (sometimes called the 'central region') and the chromosome axis.

In the nematode *C. elegans,* the SC comprised of at least four structural proteins (SYP-1–4), which are mutually dependent for their chromosome association (*MacQueen et al., 2002*; *Colaiácovo et al., 2003*; *Smolikov et al., 2007, 2009*; *Schild-Prüfert et al., 2011*). When the chromosome axis is disrupted — for example, in *C. elegans* mutants lacking either the essential meiotic axis protein HTP-3, or all three meiotic kleisins (REC-8, COH-3, and COH-4) — SC proteins fail to assemble between chromosomes, and instead form large bodies in the nucleoplasm of meiotic cells (*Figure 1A* and *Figure 1—figure supplement 1A*) (*Goodyer et al., 2008*; *Severson et al., 2009*). Using transmission electron microscopy (TEM), we found that these nuclear bodies have a multilaminar internal structure, in which each layer recapitulates the appearance and dimensions of SCs (*Figure 1B* and *Figure 1—figure supplement 2A and C*) (*Dernburg et al., 1998*; *Schild-Prüfert et al., 2011*). Similar periodic, multilaminar structures formed by SC proteins have been observed in wild-type nuclei or cytoplasm of meiocytes or nurse cells from a wide variety of organisms, as well as under perturbed conditions where SCs cannot assemble between chromosomes, and are known as 'polycomplexes' (*Roth, 1966*; *Westergaard and von Wettstein, 1972*; *Page and Hawley, 2004*). In *C. elegans* these structured bodies have also been observed in wild-type meiocytes prior to SC assembly (*Goldstein, 2013*; *Rog and Dernburg, 2015*). Importantly, polycomplexes in *htp-3* mutants do not contain any of the known chromosome axis components (cohesins, HIM-3, or HTP-1–3) (*Severson and Meyer, 2014*, and our observations), indicating that these proteins are not essential for self-assembly of SC proteins into an ordered structure, but only for their assembly as a unilamellar structure between chromosomes. Additionally, this reveals that the electron-dark bands with 100 nm periodicity within polycomplexes, which correspond to the cytologically-defined 'lateral elements' of the SC, are not equivalent to chromosome axes.

Polycomplexes contain all known SC structural proteins and depend on the full cohort of these proteins for their formation (*Humphryes et al., 2013*, and our observations). Coupled with their structural similarity to SCs, we reasoned that it might be possible to learn more about the SC, independent of its interactions with chromosome axes, by analyzing polycomplex dynamics in living worms. Time-lapse imaging of *htp-3* mutant

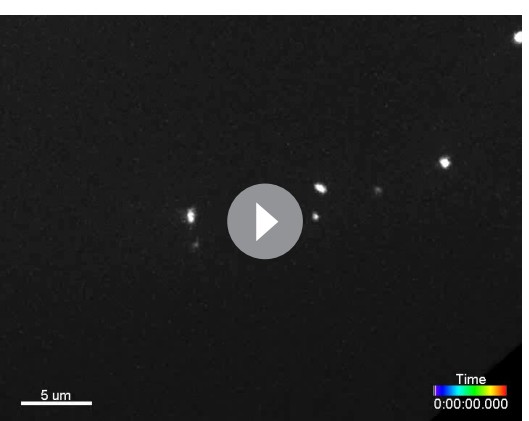

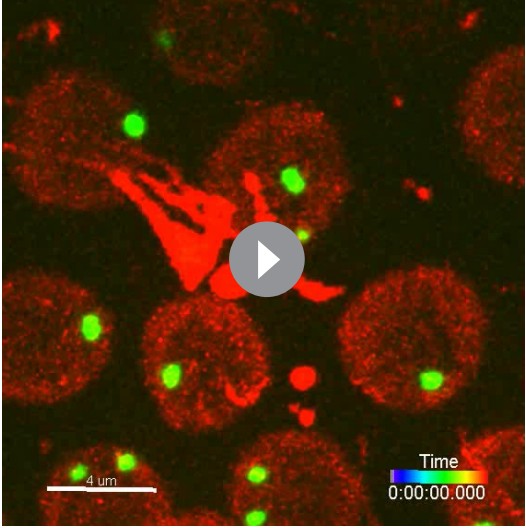

**Video 1.** Polycomplexes fuse upon contacting each other. Oocytes from the diplotene region of an *htp-3 (tm3655)* hermaphrodite expressing GFP-SYP-3, corresponding to the stills shown in *Figure 1C*. Polycomplexes exhibit constant deformations and merge with one another (between t = 0:09:00 and t = 0:10:00 and between t = 0:20:00 and t = 0:21:00). Meiotic progression in this recording is right to left. Images were acquired every 1 min. Playback is 300x real-time. Scale bar = 5 μm..

**Video 2.** Polycomplexes merge upon contact. Pachytene nuclei from an intact *htp-3(tm3655)* hermaphrodite expressing GFP-SYP-3 and histone-mCherry, corresponding to the stills in *Figure 1—figure supplement 2B*. Polycomplexes (green) merged with one another (between t = 0:40:04 and t = 0:42:44). An apoptotic nucleus was being engulfed between t = 1:24:05 and t = 1:25:25. A polycomplex remains visible until t = 1:26:45. Meiotic progression in this recording is from left to right. Images were acquired about every 1 min. Playback is 400x real-time. Scale bar = 4 μm.

worms expressing GFP-SYP-3 revealed that prophase nuclei frequently contained more than one fluorescent polycomplex. Unexpectedly, we observed that when these came into contact, they readily fused to form a single larger body (*Videos 1* and *2*, *Figure 1C* and *Figure 1—figure supplement 2B*). We also observed polycomplexes to undergo large-scale structural deformations, particularly towards the end of prophase as the nuclear volume increased (*Video 1* and *Figure 1C*). The ability of such bodies to merge together and to readily change in shape are defining characteristics of liquids, since they imply that molecules can rapidly rearrange within a material. These observations were thus initially very surprising, given the highly ordered appearance of polycomplexes. However, materials with liquid-like properties and ordered arrangements of molecules, known as structured fluids or liquid crystals, are a well-known class of soft condensed matter, and a variety of structures within cells have been postulated to have such properties (*Brown and Wolken, 1979*; *Rey, 2010*).

In vivo imaging also revealed that polycomplexes likely assemble by regulated phase separation. Our early efforts to carry out long-term recordings in living worms consistently led to meiotic arrest, which we deduced to be a physiological response to food withdrawal (*Wynne et al., 2012*; *Rog and Dernburg, 2015*). Arrest was evident as abatement of the dramatic chromosome movements that normally occur throughout early prophase (*Wynne et al., 2012*), which typically ceased within 15–20 min of immobilization (*Figure 2A* and *Figure 2—figure supplement 1* and *Videos 3* and *4*; see *Figure 2B* for a schematic of the *C. elegans* gonad). In nuclei that had initiated synapsis before arresting, we observed that diffuse GFP-SYP-3 gradually coalesced to join existing segments of SC between chromosomes. These partial SCs became brighter but not longer, likely reflecting addition of subunits to form a multilayered structure, much as they must stack within 3D polycomplexes. Consistent with this idea, EM analysis of lateral and cross-sectional views of SCs has indicated that they contain varying numbers of layers (*Schmekel et al., 1993a*, *1993b*). The observation that these pre-existing SC segments did not spread along partially synapsed chromosomes following arrest suggests that longitudinal extension is an active process. In earlier, presynaptic nuclei, GFP-SYP-3 coalesced within each nucleus over a period of tens of minutes to form several compact aggregates. Similar coalescence of SC proteins was observed in animals treated with sodium azide, which reversibly arrests meiosis and other physiological processes by depleting ATP (*Figure 2C*), suggesting that phosphorylation and/or other ATP-dependent mechanisms normally maintain the solubility of SC subunits.

The liquid-like behaviors we observe for polycomplexes, as well as the spontaneous coalescence of SC components from the nucleoplasm upon arrest, indicate that these nuclear bodies arise through coacervation, or phase separation, as been inferred for other self-assembling subcellular compartments (*Brangwynne et al., 2009*; *Hyman et al., 2014*). SC proteins have also been shown to form nuclear aggregates in *C. elegans* following extended incubation at elevated temperatures, which also renders the animals sterile (*Bilgir et al., 2013*). We found that these heat-induced bodies lack periodic striations in electron micrographs and do not undergo fusions or shape deformations (*Figure 1—figure supplement 1B–D* and *Video 5*), indicating that they lack both the liquid-like properties and internal order of polycomplexes. This irreversible aggregation likely represents conversion to a denatured state, similar to the way that other coacervates can transition to aberrant solid or amyloid-like states (*Kroschwald et al., 2015*).

We next investigated whether SCs assembled between chromosomes also exhibit liquid-like behaviors by assessing the mobility of SC proteins within these structures. We engineered worm strains expressing SYP-3 fused to the photoconvertible fluorescent protein mMaple3 (*Wang et al., 2014b*). We photoconverted subnuclear regions in live worms and imaged these nuclei over time. The photoconverted proteins rapidly redistributed throughout the SCs within the same nucleus, becoming homogeneously dispersed within 22 min (*Figure 3A,C* and *Figure 3—figure supplement 1A*). By contrast, when the axis protein HIM-3 was tagged with mMaple3, we observed no significant redistribution of converted fluorophores over more than 30 min of observation (*Figure 3B–C* and *Figure 3—figure supplement 1B*). This implies that other axis components, particularly HTP-3, which scaffolds the recruitment of HIM-3 (*Kim et al., 2014*), are also stably associated with the axis. We conclude that SC central region proteins are far more mobile than axis components. The relatively slow rate of subunit turnover within the SC compared to that observed for other liquid-like compartments might reflect a high viscosity of the liquid-like phase, and/or other constraints such as confinement within the very thin laminar space, or a slow rate of exchange of subunits between the different SC compartments in the nucleus. Future studies using high-speed imaging and other tools

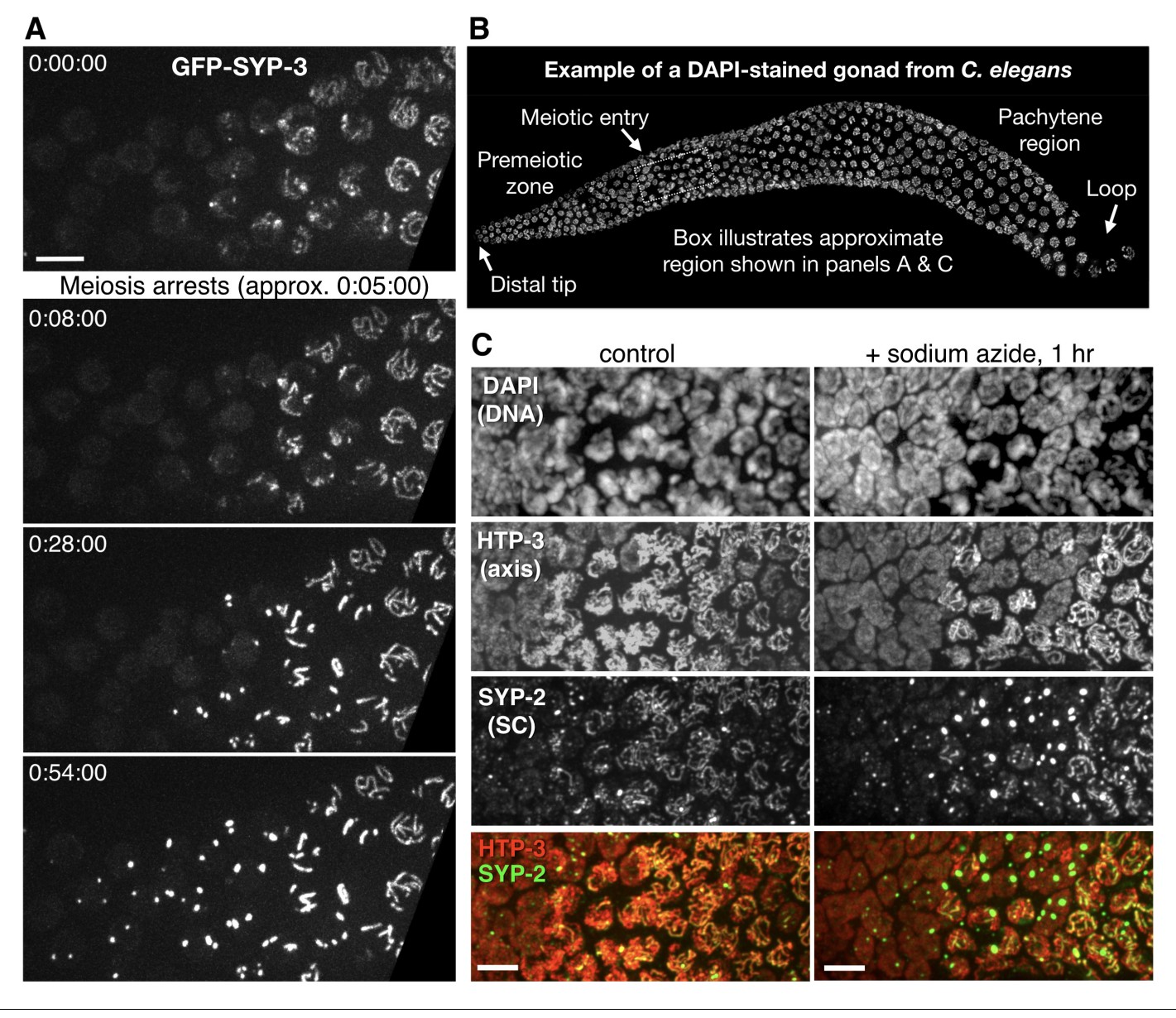

**Figure 2.** Condensation of SC proteins following meiotic arrest or ATP depletion. (**A**) Still images from a time-lapse recording of an otherwise wild-type hermaphrodite expressing GFP-SYP-3, sampled at 1 min intervals. A portion of the germline containing early meiotic nuclei, corresponding approximately to the boxed region in panel (**B**), is shown. During this recording, chromosome movements abated at approximately 0:05:00. Following this arrest, gradual condensation of fluorescent bodies is observed in nuclei lacking preexisting stretches of SC (left side of frame), while existing SC segments become brighter, but do not elongate (near the right side of frame). Scale bar = 5 μm. The full recording is shown in *Video 3*. (**B**) The proximal region of a gonad from a *C. elegans* hermaphrodite, fixed and stained with DAPI, is shown as a reference for other figures and recordings. Each gonad contains a complete progression of meiotic stages; here, nuclei enter and advance through meiotic prophase from left to right. A signal from the distal tip inhibits meiotic entry; once cells move far enough from this signal they enter meiosis and progress through prophase. At the loop region of the gonad, a spatially regulated signal triggers exit from pachytene. (**C**) Projection images from deconvolved 3D image stacks, showing premeiotic and early meiotic nuclei from wild-type adult hermaphrodites that were incubated for 1 hr on plates containing 0.5% w/v sodium azide before fixation, and control (non-azide-treated) animals. Gonads were dissected and fixed immediately following azide treatment, and stained with antibodies against SYP-2 and HTP-3 to localize SCs and chromosome axes, respectively. Small polycomplexes that contain SYP-2 but not HTP-3 are observed in some early meiotic nuclei in the absence of any treatment, as previously described (*Goldstein, 2013*), but become much larger and more abundant in response to azide exposure. Note that transition zone nuclei maintain their polarized chromatin morphology during arrest. Scale bars = 5 μm.

*Figure 2 continued on next page*

*Figure 2 continued*

The following figure supplement is available for figure 2:

**Figure supplement 1.** Condensation of polycomplexes from the nucleoplasm following meiotic arrest.

should provide a more complete understanding of the movement of proteins within the SC. Taken together, our observation of coacervation following meiotic arrest, of liquid-like behaviors of polycomplexes, and of the dynamic exchange of subunits all indicate that the SC is a phase-separated compartment with liquid crystalline properties.

## SC assembly depends on on hydrophobic and electrostatic interactions

Recent work has illuminated several mechanisms that can promote intracellular phase transitions, including multivalent interactions (*Li et al., 2012*), low-complexity protein domains with β-strand propensity (*Kato et al., 2012*), and hydrophobic interactions (*Schmidt and Görlich, 2015*). The latter are thought to be essential for the integrity of the nuclear pore, within which a network of unstructured proteins forms a diffusion barrier, and for the formation of P-granules in the *C. elegans* germline (*Updike et al., 2011*), among other liquid-like cellular compartments (*Kroschwald et al., 2015*). In support of this idea, these compartments are reversibly disrupted by moderate concentrations (~5%) of 1,6-hexanediol or similar aliphatic alcohols, which lower aqueous surface tension (*Romero et al., 2007*) and reduce the hydrophobic effect (*Ribbeck and Görlich, 2002*).

To test the idea that hydrophobic interactions promote coacervation of SC proteins, we extruded worm gonads in physiological buffer and added varying concentrations of 1,6-hexanediol. When gonads expressing GFP-SYP-3 were exposed to 5–10% 1,6-hexanediol, we observed immediate disruption of SCs and dispersion of the fluorescence throughout the nucleoplasm. By contrast, the distributions of many other chromosome-associated proteins were unaffected by this treatment, including axis components (HTP-3, HIM-3, HTP-1/2, LAB-1, and cohesins); the pairing center protein HIM-8; and proteins associated with CO sites (COSA-1 and ZHP-3) (*Figure 4A–B*, *Figure 4—figure supplement 3A*, *Videos 6* and *7*, and data not shown; see *Figure 4—figure supplement 1C* for an explanatory diagram; in all figures, panels showing samples treated with 1,6-hexanediol have pink borders). In some nuclei a short stretch of SC associated with the *X* chromosomes persisted after treatment with 1,6-hexanediol (*Figure 4—figure supplement 3*), perhaps reflecting enhanced recruitment of SC proteins to the paired sex chromosomes (*Couteau et al., 2004*; *Hayashi et al., 2010*). SC dissolution following 1,6-hexanediol exposure was also observed in wild-type gonads, indicating that it was not a consequence of the fluorescent tag (*Figure 4—figure supplement 3A*). Polycomplexes showed a very similar response to 1,6-hexanediol: they rapidly dissolved upon exposure to the solvent, while heat-induced SC aggregates were resistant to dissolution (*Figure 5B*, *Figure 4—figure supplement 4* and *Video 8* and *Video 9*).

Despite the widespread conservation of SC structure among eukaryotes, central region proteins are notorious for their divergence in primary sequence among different lineages (*Westergaard and von Wettstein, 1972*; *Page and Hawley, 2004*; *Fraune et al., 2012*). We therefore wondered whether the biophysical properties we observed in *C. elegans* are conserved. We found that SCs in both *Drosophila* and budding yeast were rapidly dispersed by 1,6-hexanediol, while axis markers (Rec8p in budding yeast; C(2)M in *Drosophila*) remained associated with chromosomes, as in *C. elegans* (*Figure 4C–E*; see *Figure 4—figure supplement 1C* for an explanatory diagram). This indicates that weak hydrophobic interactions are important for SC integrity in diverse lineages.

While SCs in budding yeast, *Drosophila*, and *C. elegans* meiocytes were readily disrupted by 1,6-hexanediol, we found that similar concentrations of the more polar solvents ethanol or dimethyl sulfoxide failed to dissolve these structures (data not shown). Because the effects of 1,6-hexanediol on cellular structures are not well understood, we also probed the sensitivity of SCs to disruption by a number of other water-miscible alcohols. These di-alcohols vary in the length of the hydrocarbon chain and the positions of their hydroxyl groups. Most experiments were performed with yeast cells arrested in mid-pachytene, since it is simple to manipulate the media in which these cells are immersed and no dissection or cell permeabilization was required to see the effects of 1,6-hexanediol. For each di-alcohol we determined the minimal concentration required for complete SC

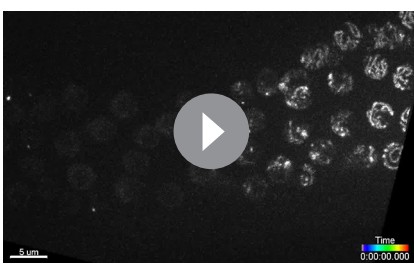

**Video 3.** Aggregation of SC components following cessation of meiotic chromosome movements. Meiotic prophase in a hermaphrodite expressing GFP-SYP-3. Premeiotic nuclei are at the left, and nuclei enter meiosis and progress from left to right. Chromosome motion ceases, indicative of meiotic arrest, at about t = 5 min. Images were acquired every 1 min. Playback speed is 300x real-time. Scale bar = 5 μm. Stills are shown in *Figure 2A*.

disruption, based on loss of any discernible structure by fluorescence microscopy of unfixed yeast cells (*Figure 4—figure supplement 1B*). We observed a clear correlation between the extent of the hydrophobic domains of different solvents and their effects on the SC. For example, 1,2-heptanediol disrupted the structure at lower concentrations than 1,2-hexanediol, which was more potent than 1,2-pentanediol. Similar results were obtained for the series 1,7-heptanediol, 1,6-hexanediol, and 1,5-pentanediol. Together, these results reinforce the idea that solvation by aliphatic alcohols is due to suppression of the hydrophobic effect, and support a major contribution of hydrophobic interactions to SC stability.

We also probed the effect of electrostatic interactions on SC stability. We noted that SC central region proteins in various organisms are predominantly polyampholytic, *i.e.*, they contain an unusually large fraction of charged residues, both acidic and basic, which are well-mixed within the primary sequences. This suggests that electrostatic interactions likely contribute to SC integrity, and that higher salt concentrations, which shield electrostatic interactions (*Veis, 2011*), might disfavor assembly. Consistent with this, we found that increasing the concentration of KCl potentiated the ability of 1,6-hexanediol to dissolve the SC in budding yeast (*Figure 4—figure supplement 1A*; other salts showed similar effects [data not shown]), highlighting a role for electrostatic interactions, in addition to hydrophobic interactions, in SC assembly and stability. The effects of increasing salt concentration were fairly modest in light of the highly charged nature of SC proteins; however, only a limited range of ionic strength could be investigated in living cells, and the effects of these buffer manipulations on intracellular ionic strength could not be directly monitored.

Both of these trends observed in budding yeast — disruption of SCs by lower

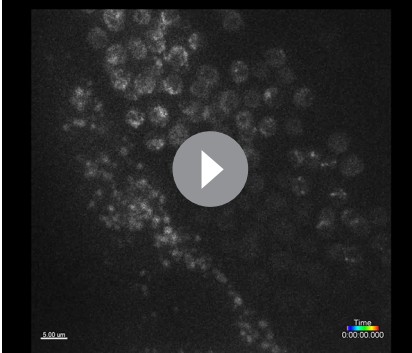

**Video 4.** Aggregation of SC components following cessation of meiotic chromosome movements. Recording from a hermaphrodite expressing GFP-SYP-3. Premeiotic nuclei are at the lower right portion of the images, and meiotic entry/progression is observed from lower right to upper central portion of the images. Fluorescent lipid droplets within the intestine are seen to the left of the gonad. Images were acquired every 1 min. Playback is 300x real-time. Scale bar = 5 μm. Stills and analysis of this recording are shown in *Figure 2—figure supplement 1*.

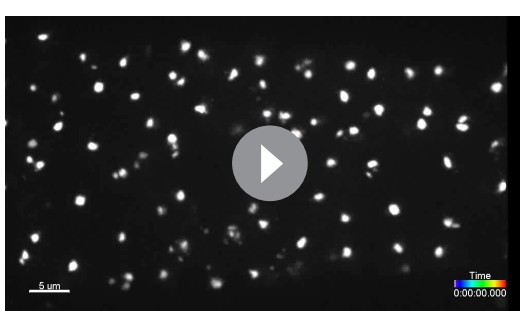

**Video 5.** Heat-induced SC aggregates do not show liquid-like properties. A *htp-3(tm3655)* mutant hermaphrodite expressing GFP-SYP-3 was incubated overnight at 26.5°C (corresponding to images in *Figure 1—figure supplement 1B*). SC aggregates show some mobility within the nucleus, but do not merge with each other, and maintain their irregular shapes over the time course. Meiotic progression in this recording is from left to right. Images were acquired every 1 min. Playback is 300x real-time. Scale bar = 5 μm.

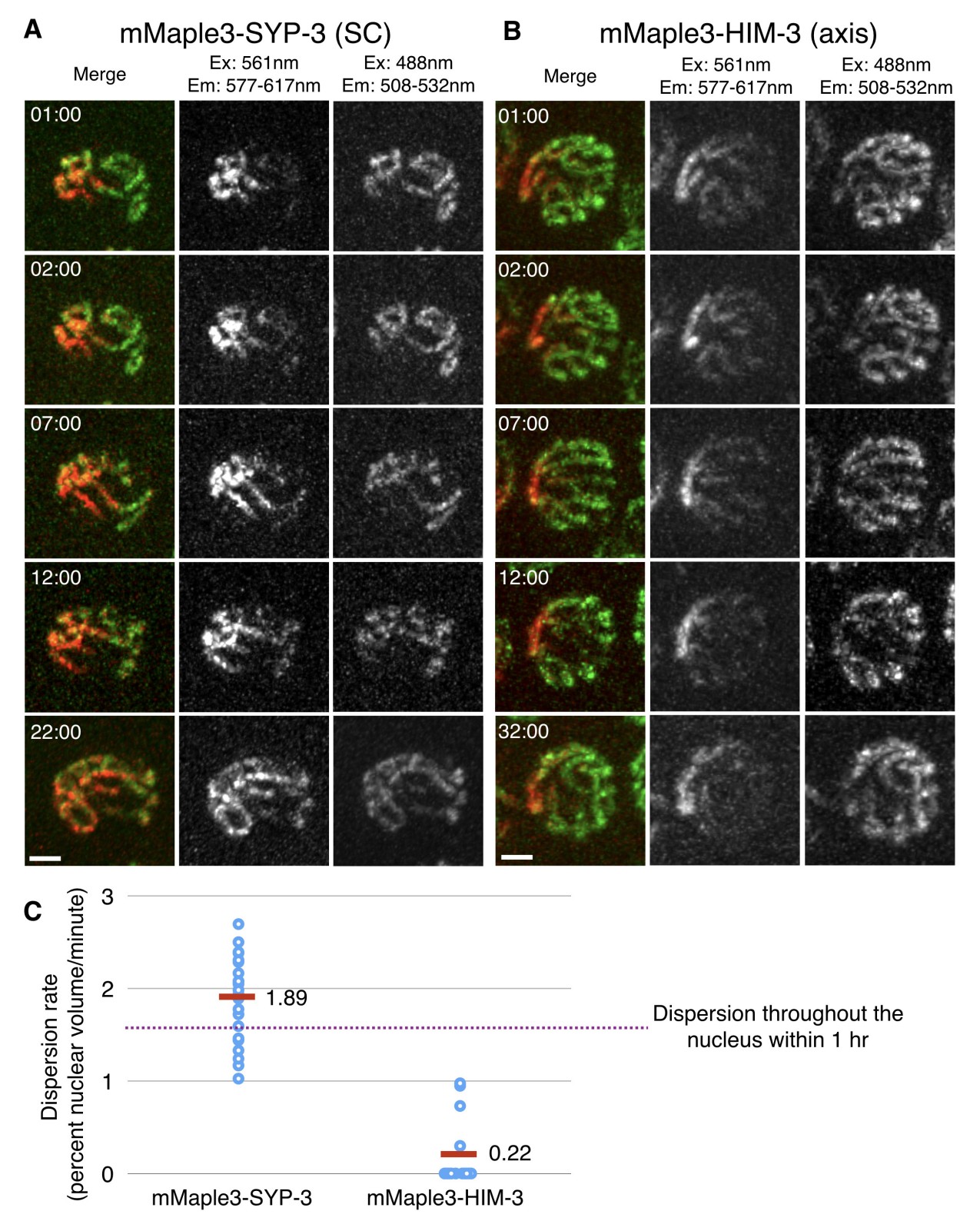

**Figure 3.** SC proteins, but not axis components, are highly dynamic. (**A–B**) Images of representative nuclei from time-lapse recordings of hermaphrodites expressing either the SC protein SYP-3 or the axis component HIM-3 fused to the photoconvertible fluorescent protein mMaple3. A subnuclear volume was photoconverted using 405 nm laser illumination at t = 00:00. The mMaple3-SYP-3 signal spread throughout the nucleus and to all chromosomes by 22 min (**A**), whereas the mMaple3-HIM-3 signal remained confined to a small region throughout the time course (32 min; **B**). Scale

*Figure 3 continued on next page*

*Figure 3 continued*

bars = 2 μm. Elapsed times are indicated as min:sec. (**C**) The mobility of HIM-3 (axis) and SYP-3 (SC) were quantified by estimating the volume of individual nuclei containing photoconverted (red) signal as a function of time (see Materials and methods for details); SYP-3 within assembled SCs is far more mobile than HIM-3 (Student's t-test: $p<10^{-13}$). The horizontal dashed line represents the expansion rate for a point source that becomes homogeneously distributed throughout the nuclear volume in one hour.

The following figure supplement is available for figure 3:

**Figure supplement 1.** SC proteins, but not axis components, are highly dynamic.

concentrations of more hydrophobic diols, and synergy between electrostatic shielding and amphipathic solvents — also held true for *C. elegans* SCs (data not shown). Experiments conducted using this system were less comprehensive since they required extensive hand dissection and, probably as a consequence of this manipulation, were somewhat less consistent and thus harder to score. Nevertheless, we tested whether various mutations in *C. elegans* that perturb meiotic recombination or chromosome structure might affect 1,6-hexanediol sensitivity, and observed no major differences, except that the aberrant SCs in *syp-3(me42)* mutants, which carry a small C-terminal truncation in an SC protein (*Smolikov et al., 2007*), were hypersensitive to 1,6-hexanediol (*Figure 4—figure supplement 2A–B*). We further probed the effects of this mutation, and found that the *syp-3(me42)* truncation also prevents polycomplex formation in an *htp-3* background (*Figure 4—figure supplement 2C*), as well as heat-induced aggregation (*Figure 4—figure supplement 2D*). The correlation between 1,6-hexanediol hyper-sensitivity, defects in SC assembly and lack of SC aggregation further underscore the contribution of hydrophobic interactions to the functional integrity of the SC.

When SCs in *C. elegans* were disrupted by 1,6-hexanediol, followed by rapid dilution of the solvent with buffer, GFP-SYP-3 immediately relocalized along the chromosomes (*Video 10* and *Figure 5A*; throughout the figures, panels showing samples subsequent to the dilution are shown with a yellow border). Some fluorescent material that had exited the nucleus, perhaps due to transient disruption of the nuclear pores, formed small puncta in the cytoplasm. Notably, reassembly of SCs following 1,6-hexanediol dilution occurred much more quickly than physiological SC assembly between axes, which requires 20–30 min per chromosome pair (*Rog and Dernburg, 2015*). Polycomplex dissolution was also reversible: upon dilution of 1,6-hexanediol, multiple small puncta formed in each nucleus, and rapidly fused with each other to form larger bodies (*Video 9* and *Figure 5B*), qualitatively similar to the behavior of polycomplexes in live animals (*Figure 1C*, *Figure 1—figure supplement 2B* and *Videos 1* and *2*). This reassembly following hexanediol dilution occurred much more quickly than under metabolic arrest (*Figure 2A* and *Figure 2—figure supplement 1*). Dissolution of the SC by 1,6-hexanediol in budding yeast cells was also reversible: we observed some re-association of SC components with chromosomes after both short (2 min) and longer (>30 min) incubations of yeast cells arrested at mid-pachytene (*Figure 5C*). These results provide additional evidence that neither the chromosome axes, which are essential for SC assembly between chromosomes, nor the SC proteins themselves, are irreversibly perturbed by exposure to 1,6-hexanediol.

## Spatial regulation of recombination proteins by the SC

Subcellular liquid-like compartments are selectively permeable to macromolecules, and can regulate biochemical reactions by concentrating enzymes and substrates and restricting their diffusion (reviewed by *Hyman et al., 2014*). We therefore wondered whether the SC might regulate and surveil recombination by partitioning CO factors into a distinct compartment between homologous chromosomes. At least one essential CO factor that is not required for SC assembly, the conserved RING-finger protein ZHP-3, is observed throughout the SC during early prophase, before becoming restricted to designated CO sites (*Figure 6—figure supplement 1A*) (*Jantsch et al., 2004*; *Bhalla et al., 2008*), whereupon it colocalizes with COSA-1, another conserved protein required for COs (*Yokoo et al., 2012*). We found that CO formation altered not only the localization of ZHP-3 but also its sensitivity to 1,6-hexanediol: Prior to CO designation, the ZHP-3 distributed throughout SCs was dispersed by the solvent, while CO-associated ZHP-3 foci remained intact even after disruption of the SC by 1,6-hexanediol (*Figure 6—figure supplement 1A–B*).

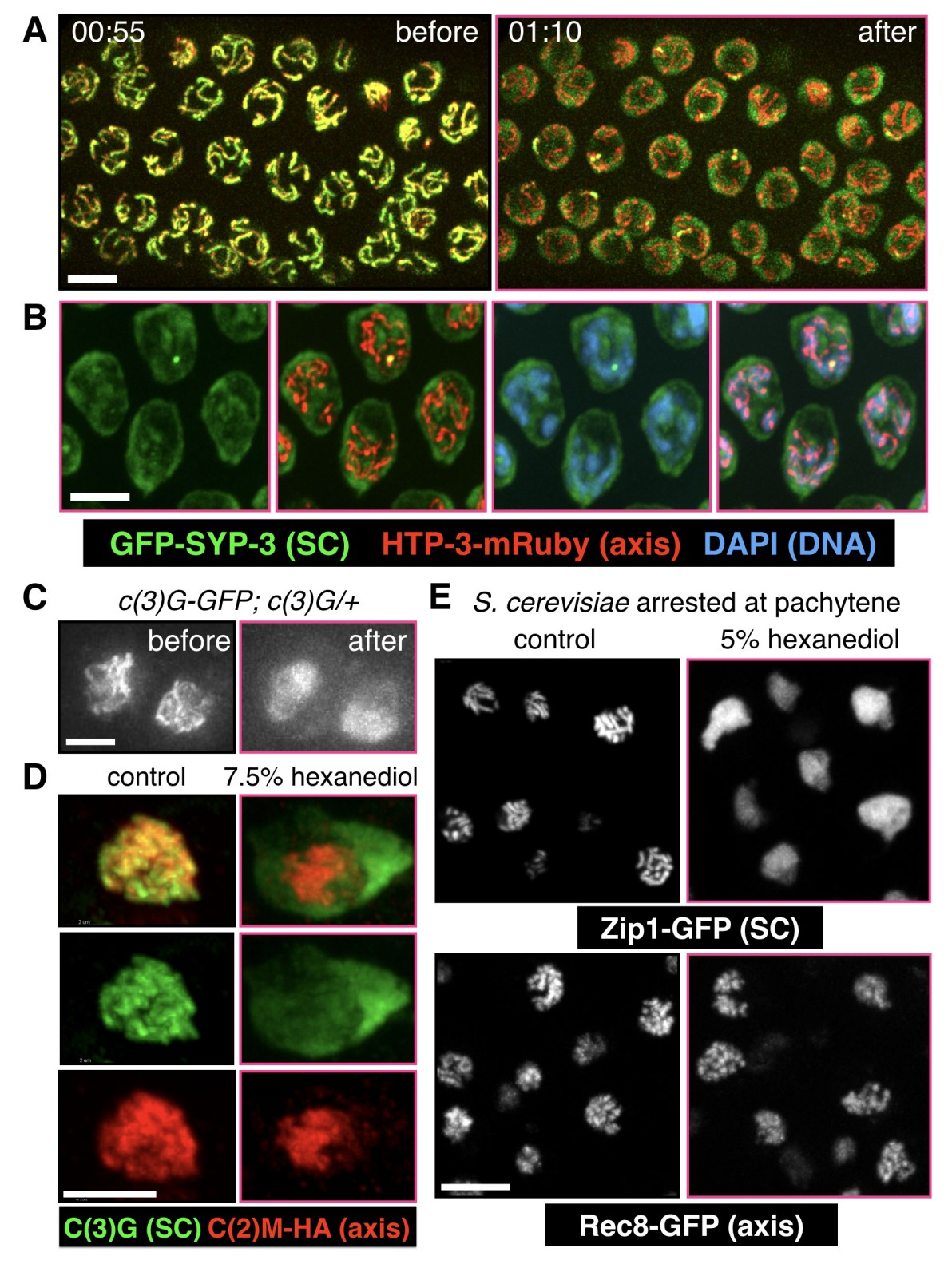

**GFP-SYP-3 (SC)   HTP-3-mRuby (axis)   DAPI (DNA)**

**Figure 4.** SCs in *C.elegans, S. cerevisiae* and *Drosophila melanogaster* dissolve in the presence of 1,6-hexanediol. (**A**) Projection images showing selected time points from time-lapse recordings of extruded gonads from *C. elegans* hermaphrodites expressing GFP-SYP-3 and HTP-3-mRuby to mark SCs and axes, respectively. Upon exposure to 5% 1,6-hexanediol, SYP-3 immediately dispersed throughout the nucleoplasm, while HTP-3 remained associated with chromosomes. Most of the GFP-SYP-3 fluorescence remained in the nucleus for the duration of our experiments. The full recording is
*Figure 4 continued on next page*

*Figure 4 continued*

shown in **Video 6**. Scale bar = 5 µm. (B) Gonads were treated with 1,6-hexanediol as in (A), but then fixed and counterstained with DAPI. Scale bar = 5 µm. (C–D) The SC in *Drosophila* is also sensitive to 1,6-hexanediol. (C) Projection images showing GFP fluorescence in two adjacent oocytes from a *Drosophila* female expressing GFP-C(3)G. C(3)G is an essential component of the SC central region. Ovarioles were dissected in buffer, and imaged before and after exposure to 7.5% 1,6-hexanediol. Scale bar = 5 µm. (D) Projection images showing a single oocyte within a germarium from a fly expressing HA-tagged C(2)M (a component of the chromosome axis). Ovarioles were dissected, exposed to 7.5% hexanediol (or not), and fixed with formaldehyde, then stained with antibodies against C(3)G (green) and HA (red). C(3)G was solubilized by hexanediol, while C(2)M remained associated with chromosome axes. Similar results were obtained with flies expressing GFP-C(3)G and stained with antibodies against GFP and C(2)M (data not shown). Scale bar = 5 µm. (E) Diploid *ndt80 S. cerevisiae* strains expressing the indicated GFP fusion proteins were sporulated and arrested in mid-pachytene. Following exposure to 1,6-hexanediol, cells were imaged immediately without fixation. The central region component Zip1p dispersed throughout the nuclei upon 1,6-hexanediol treatment, while the cohesin subunit Rec8p remained along chromosomes. Scale bar = 5 µm.

The following figure supplements are available for figure 4:

**Figure supplement 1.** SC proteins, but not axis components, are highly dynamic.

**Figure supplement 2.** The *syp-3(me42)* mutation results in hypersensitivity of SCs to 1,6-hexanediol and prevents aggregation in the absence of axes or at high temperatures.

**Figure supplement 3.** The *X* chromosome pairing center protein HIM-8 and a some of the SC on the *X* chromosome are refractory to 1,6-hexanediol.

**Figure supplement 4.** Polycomplexes, but not heat-induced aggregates, are dissolved by 1,6-hexanediol.

---

If ZHP-3 concentrates between chromosomes by partitioning into the SC, we expected that it might also localize within polycomplexes. Indeed, we found that in *htp-3* mutants, ZHP-3 was concentrated throughout polycomplexes during most of meiotic prophase. However, near the 'loop' region of the gonad, where exit from pachytene normally occurs (see *Figure 2B* for reference), we observed an abrupt change in ZHP-3 localization: The protein largely disappeared from the interior of the polycomplexes and became restricted to a single focus conspicuously abutting the surface of each polycomplex. Intriguingly, at the same stage, COSA-1, which was not detected earlier, became localized to these same small, peripheral puncta (*Figure 6A–C*). This dynamic relocalization of ZHP-3 and COSA-1 is highly analogous to what is observed along bona fide SCs upon CO designation (*Yokoo et al., 2012*). We further determined that ZHP-3 is required for the appearance of COSA-1 foci at polycomplexes, while COSA-1 is required for the relocalization of ZHP-3 to the edge of these bodies and its disappearance from the interior (*Figure 6D–E* and *Figure 6—figure supplement 1C–D*), mirroring the interdependence of these factors for their localization to CO sites (*Yokoo et al., 2012*). We also found that the earlier ZHP-3 localization throughout polycomplexes is readily disrupted by 1,6-hexanediol, whereas the foci of COSA-1 and ZHP-3 foci at the edges of polycomplexes are 1,6-hexanediol-resistant (*Figure 6—figure supplement 1E*), again mirroring the properties of ZHP-3 and COSA-1 associated with SCs and designated CO sites, respectively. Polycomplexes do not appear to be associated with chromosomes and moreover, meiotic recombination does not initiate in *htp-3* mutants (*Goodyer et al., 2008*). Thus, localization of COSA-1 and ZHP-3 to foci on polycomplexes does not require their interaction with CO intermediates such as Holliday junctions, or indeed with chromosomes. These results imply that in response to exogenous cell cycle signals, polycomplexes can recapitulate a signaling network that normally designates COs, and hence suggest how a liquid-like SC compartment may promote and regulate CO recombination along meiotic chromosomes.

## Discussion

Sixty years of observation by EM has revealed that SC structure, as well as the tendency of its components to assemble into ordered polycomplexes, are widely conserved (*Roth, 1966*; *Westergaard and von Wettstein, 1972*; *Page and Hawley, 2004*). Here we report another conserved property of SCs and polycomplexes: they assemble through coacervation of polypeptides into compartments with liquid crystalline properties. While the symmetric, transversely striated appearance of the SC, combined with known properties of coiled-coil proteins, has suggested a

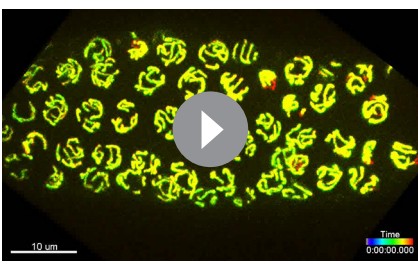

**Video 6.** The central region of the SC, but not the chromosome axis, is dissolved by 1,6-hexanediol. This gonad was extruded from a hermaphrodite expressing GFP-SYP-3 and HTP-3-mRuby. Partially synapsed zygotene nuclei are seen at the left, and fully synapsed pachytene nuclei at the right. 1,6-hexanediol was added to a concentration of 5% at t = 0:00:20. Images were acquired every 5 s. Playback is 50x real-time. Scale bar = 5 μm. Stills are shown in *Figure 4A*.

filamentous and stable internal structure, our observations indicate that SCs instead resemble smectic (planar) liquid crystal bilayers, structurally analogous to lipid bilayers. Moreover, the SC can also assemble into its canonical smectic structure when its normal association with chromosome axes is prevented. The preferential assembly of SC proteins into a thin lamellar structure between two axes is further evidenced by our observation that upon meiotic arrest, nuclear aggregates appear only if no pre-assembled SC bilayers are present (*Figure 2*). While biological membranes are well-known examples of smectic liquid crystals, to our knowledge, the SC is the first known example of such a highly ordered, liquid-like structure arising from protein-protein interactions. However, a variety of biological assemblies of less regular structure have been described as liquid crystals (*Brown and Wolken, 1979*; *Rey, 2010*), and many others have been created in vitro by mixing synthetic peptides in aqueous solution.

Liquid crystals are considered to be a distinct phase of matter from liquids, which are disordered and anisotropic. Thus, it is perhaps unsurprising that the proteins that assemble to form SCs are markedly different from those that have been implicated in the assembly of other liquid-like compartments within cells. Many such proteins have low-complexity, intrinsically disordered domains that are essential for their phase transition behavior (*Hyman et al., 2014*; *Zhu and Brangwynne, 2015*), and some of these have been shown to adopt cross-beta strand structures in vitro (*Kato et al., 2012*). By contrast, SC proteins are predicted to be largely alpha-helical in structure, and have extensive domains predicted to form coiled-coils. Smectic liquid crystals assemble from elongated, amphipathic rod-like molecules (*Gennes and Prost, 1993*). It is known that some of the longer SC proteins, particularly SYCP1 in mammals and Zip1p in budding yeast, have a head-to-head orientation within the SC, with their N-termini near the center and their C-termini near the chromosome axes (*Liu et al., 1996*; *Schmekel et al., 1996*; *Dong and Roeder, 2000*; *Anderson et al., 2005*). Interaction of their C-termini with axis-associated proteins seems to promote their coacervation with other SC proteins, possibly by promoting high local concentrations. Axis association also favors the formation of single bilayers (lamellae), rather than the stacks of bilayers that make up polycomplexes. By specifying the orientation of these 'transverse filament' proteins, the chromosome axes may also act as a 'director' for the liquid crystalline array.

Many of the best-studied coiled-coil proteins form highly stable interactions through regions of superhelical coiling. Some, such as muscle myosin, keratins, and lamins, form long-lived filamentous networks, which has suggested that the SC might have similar properties. However, it has long been appreciated that interruptions in the phasing of the canonical heptad repeat sequence disrupt supercoiling and reduce the stability of interactions (*Brown et al., 1996*). We note that even the 'transverse filament' proteins SYCP1 (vertebrates and other metazoans), Zip1p (budding yeast), C(3)G (*Drosophila*), and SYP-1 (*Caenorhabditis*) (*Page and Hawley, 2004*)

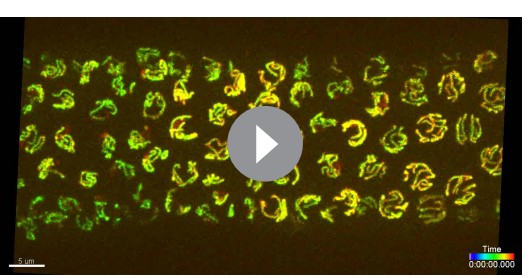

**Video 7.** 1,6-hexanediol dissolves the SC central region. Recording from a gonad extruded from a hermaphrodite expressing GFP-SYP-3 and HTP-3-mRuby. 5% 1,6-hexanediol was added at t = 0:00:35 (yellow flash). Meiotic progression in this recording is from right to left. Images were acquired every 5 s. Playback is 25x real-time. Scale bar = 10 μm.

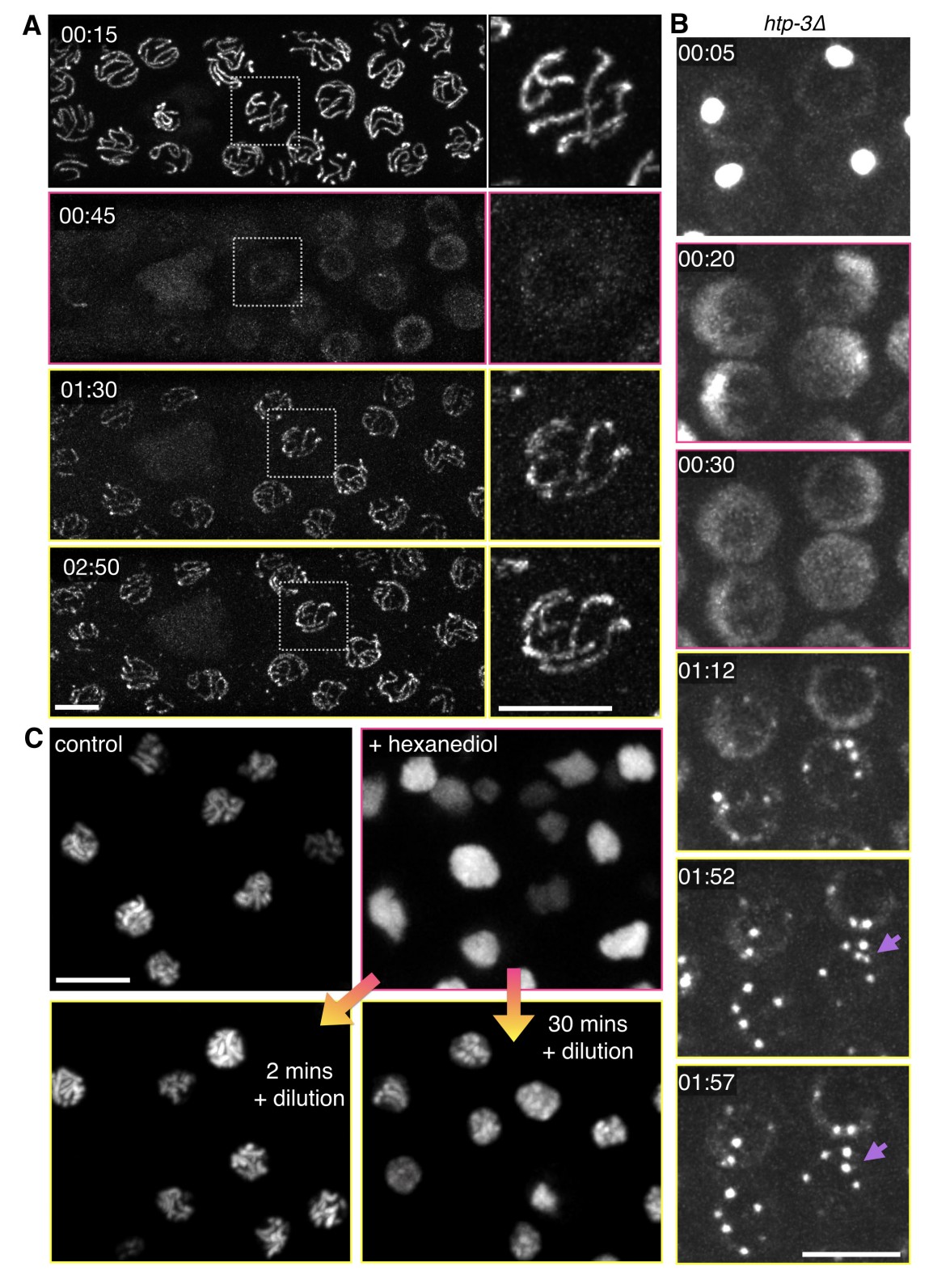

**Figure 5.** Dissolution of SCs and polycomplexes by 1,6-hexanediol is reversible. (**A**) Projection images from a time-lapse recording showing GFP-SYP-3 before and after dispersion by 5% 1,6-hexanediol, followed by dilution with buffer. Pink borders around images indicate times after hexanediol addition, and yellow borders indicate times after dilution of the hexanediol with buffer. Upon dilution, fluorescence reappears along chromosomes. Some small puncta are also observed in the cytoplasm. Chromosome association of SC proteins after longer exposures to 1,6-hexanediol (>2 min) was

*Figure 5 continued on next page*

*Figure 5 continued*
not reversible (data not shown), perhaps due to irreversible perturbations of chromosome structure or SC subunits. The full recording is shown in
*Video 9*. Elapsed time is indicated as mm:ss. Larger-magnification images of a representative nucleus are shown on the right. Scale bars = 5 μm. (B)
Projection images from a time-lapse recording of a *htp-3(tm3655)* mutant hermaphrodite expressing GFP-SYP-3. Exposure to 7.5% 1,6-hexanediol (pink
image borders) induces rapid dispersion of the SC proteins from polycomplexes. Upon dilution of 1,6-hexanediol with buffer (yellow image borders),
SYP-3 coalesces into smaller bodies that fuse with each other upon contact (lilac arrows). The full recording is shown in *Video 10*. Scale bar = 5 μm. (C)
A diploid *ndt80 S. cerevisiae* strain expressing Zip1p-GFP was sporulated, arrested in mid-pachytene, and treated with 5% 1,6-hexanediol. 10 volumes
of buffer were added after the indicated incubation times with hexanediol, and cells were then mounted and imaged without fixation. Reassociation of
Zip1 with chromosomes was observed when 1,6-hexanediol was diluted after 2 min or 30 min, but appeared to be less robust after the longer
incubation. Scale bar = 5 μm.

contain very few perfect heptads arranged in tandem, which may underlie the lability of the SC.

Meiotic synapsis can be disrupted by small increases in temperature, as seen in *C. elegans* (*Bilgir et al., 2013*) and many other organisms, and we propose that this stems from the liquid crystalline properties of the SC. While the *C. elegans* Bristol N2 strain remains fertile at temperatures up to 25°C, at 26.5°C hermaphrodites are sterile, and in such animals we observed SC protein assemblies that were neither crystalline nor liquid-like. Because liquid crystals depend on ordered but weak interactions, they are considered to be a 'mesophase' between liquids and crystalline solids. They are often exquisitely sensitive to variations in temperature and/or other perturbations such as electric fields, making synthetic materials useful as thermometers (on fish tanks and mood rings) and in video displays (*Gennes and Prost, 1993*; *Carlton et al., 2013*). Thus strong selective forces to maintain the responsive, liquid-like properties of the SC, which plays a central role in fertility and fitness, may underlie the sequence diversity observed among its constituent proteins, especially among ectothermic organisms. In support of this idea, direct evidence has recently indicated that SC proteins, among other chromosome structural components, are subject to rapid positive selection in plant populations growing at different temperatures (*Wright et al., 2015*).

Our observation that meiotic arrest or azide treatment promotes the coalescence of SC components is consistent with the idea that phosphorylation and/or other modifications regulate the coacervation of these proteins. These modifications presumably tune the strength of interactions among SC proteins to control the timing and location of assembly and disassembly, as has been described for other cellular coacervation processes, such as the formation of P-granules (*Wang et al., 2014a*). Consistent with this, PLK-2, a kinase that promotes SC disassembly at the end of meiotic prophase, localizes to SCs (*Harper et al., 2011*), and we have found that several of the SYP proteins can be phosphorylated by PLK-2 in vitro (Y. Kim and AFD, unpublished). Because liquid crystals are ultrasensitive to perturbations of their intermolecular interactions (*Gennes and Prost, 1993*; *Carlton et al., 2013*), we envision that posttranslational modifications may also modulate the dynamic properties of assembled SCs during meiotic progression.

Phase-separated liquid-like compartments in vitro and in vivo are selectively and tunably permeable to macromolecules. This property can concentrate specific cellular activities while excluding others (*Hyman et al., 2014*). While the liquid crystalline properties of the SC set it apart from other compartments that resemble amorphous liquids, we provide evidence that it may have some analogous properties. Specifically, this assembly dynamically regulates the localization of at least two conserved recombination factors, ZHP-3 and COSA-1 (*Figure 6* and *Figure 6—figure supplement 1*), which are not required for SC assembly, in response to meiotic

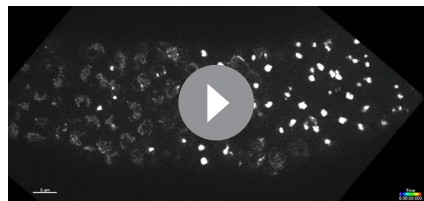

**Video 8.** Heat-induced SC aggregates are resistant to 1,6-hexanediol. Time-lapse movie of a gonad from *GFP-SYP-3* hermaphrodite incubated overnight at 26.5°C shown in *Figure 4—figure supplement 4A*. 7.5% 1,6-hexanediol was added at t = 0:00:10. SC stretches in the region of the germline at the left side of the image are quickly dissolved, while the large, irregular SC aggregates remain intact. Meiotic progression in this recording is from left to right. Images were acquired every 5 s. Playback speed is 25x real-time. Scale bar = 5 μm.

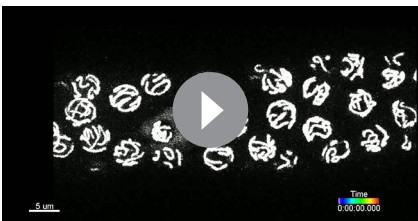

**Video 9.** 1,6-hexanediol reversibly dissolves the SC. Time-lapse movie of a gonad from *GFP-SYP-3* hermaphrodite shown in *Figure 5A*. 7.5% 1,6-hexanediol was added at t = 0:00:20, and diluted at t = 0:00:50 and 0:01:35. Meiotic progression in this recording is from right to left. Images were acquired every 5 s. Playback is 25x real-time. Scale bar = 5 µm.

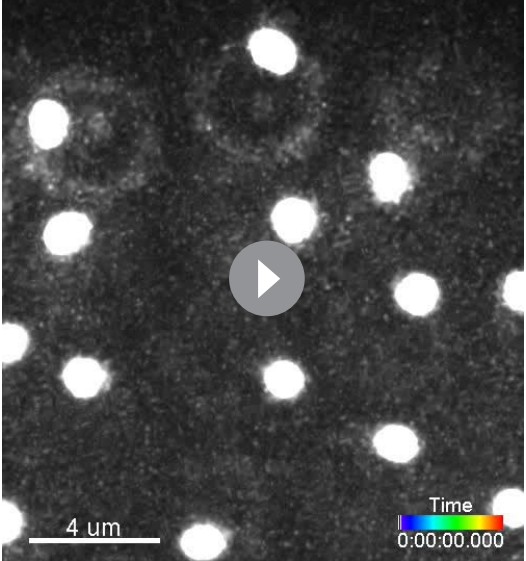

**Video 10.** 1,6-hexanediol dissolves polycomplexes. Extruded gonad from an *htp-3(tm3655)* mutant expressing GFP-SYP-3, corresponding to the stills shown in *Figure 5B*. 7.5% 1,6-hexanediol was added at t = 0:00:10, and diluted at t = 0:00:45 and t = 0:01:17. Two small polycomplexes, reformed after 1,6-hexanediol addition and dilution, merge to form a single body at t = 0:01:52. Images were acquired about every 5 s. Playback is 25x real-time. Scale bar = 4 µm.

progression. Previous work has shown that Zip3p, the budding yeast homolog of ZHP-3, also localizes throughout polycomplexes, while the CO factors Zip2p, Zip4p, Msh5p and the 9-1-1 clamp proteins localize to one or two foci abutting polycomplexes, sometimes referred to as a 'capping complex' (*Tsubouchi et al., 2006*; *Shinohara et al., 2015*), very similar to the physical distribution we observe for ZHP-3 and COSA-1 during late meiotic prophase. Homologs of ZHP-3 in plants, mammals and fungi also localize to SCs before concentrating at sites of COs (*Chelysheva et al., 2012*; *Reynolds et al., 2013*; *De Muyt et al., 2014*; *Jahns et al., 2014*; *Qiao et al., 2014*), suggesting that spatial regulation of COs by the SC is widely conserved. We envision that the SC acts as an active 'scaffold' to localize and concentrate 'client' proteins (*Banani et al., 2016*), defined here as SC-resident proteins that are not required for SC assembly. These factors, including ZHP-3, are recruited by the SC to control the processing of double-strand breaks that are induced during meiosis. We also speculate that the SC may share an evolutionary relationship with other liquid-like nuclear compartments, such as repair foci, which concentrate repair factors at sites of DNA damage (*Zhu and Brangwynne, 2015*).

Our observations support a central role for the SC in CO interference, which acts over many microns and even over entire chromosomes (*Page and Hawley, 2004*; *Libuda et al., 2013*). Evidence that SCs share physical properties with liquid crystals, a particularly responsive class of active materials, illuminates how they might contribute to the transduction of a long-range, cis-acting interference signal (*Figure 6C*). In particular, the knowledge that macromolecules can diffuse laterally within the plane of smectic liquid crystals suggests that the dynamics of diffusion within the SC likely determine the rate at which biochemical signals are propagated along the chromosomes. (By analogy, 2D diffusion within lipid membranes profoundly impacts the spatial organization and kinetics of signaling by membrane-associated proteins.) Thus, our discovery that the SC acts as a non-membrane-bound compartment suggests how COs could be regulated in cis through a reaction-diffusion mechanism, which is consistent with quantitative analysis of CO interference in several organisms (*Fujitani et al., 2002*). This idea is also consistent with evidence that reduction in the expression levels of SC proteins, which likely results in discontinuities of the SC along individual chromosomes, can permit additional COs (*Hayashi et al., 2010*; *Libuda et al., 2013*).

We further report that both SCs and polycomplexes undergo an abrupt biochemical change in response to cell cycle signals, which results in the appearance of a single CO-like focus and removal of ZHP-3 from the interior, indicating that this material can mediate switch-like regulation that spans an entire compartment. We have found that depletion of the ERK kinase MPK-1, which has been

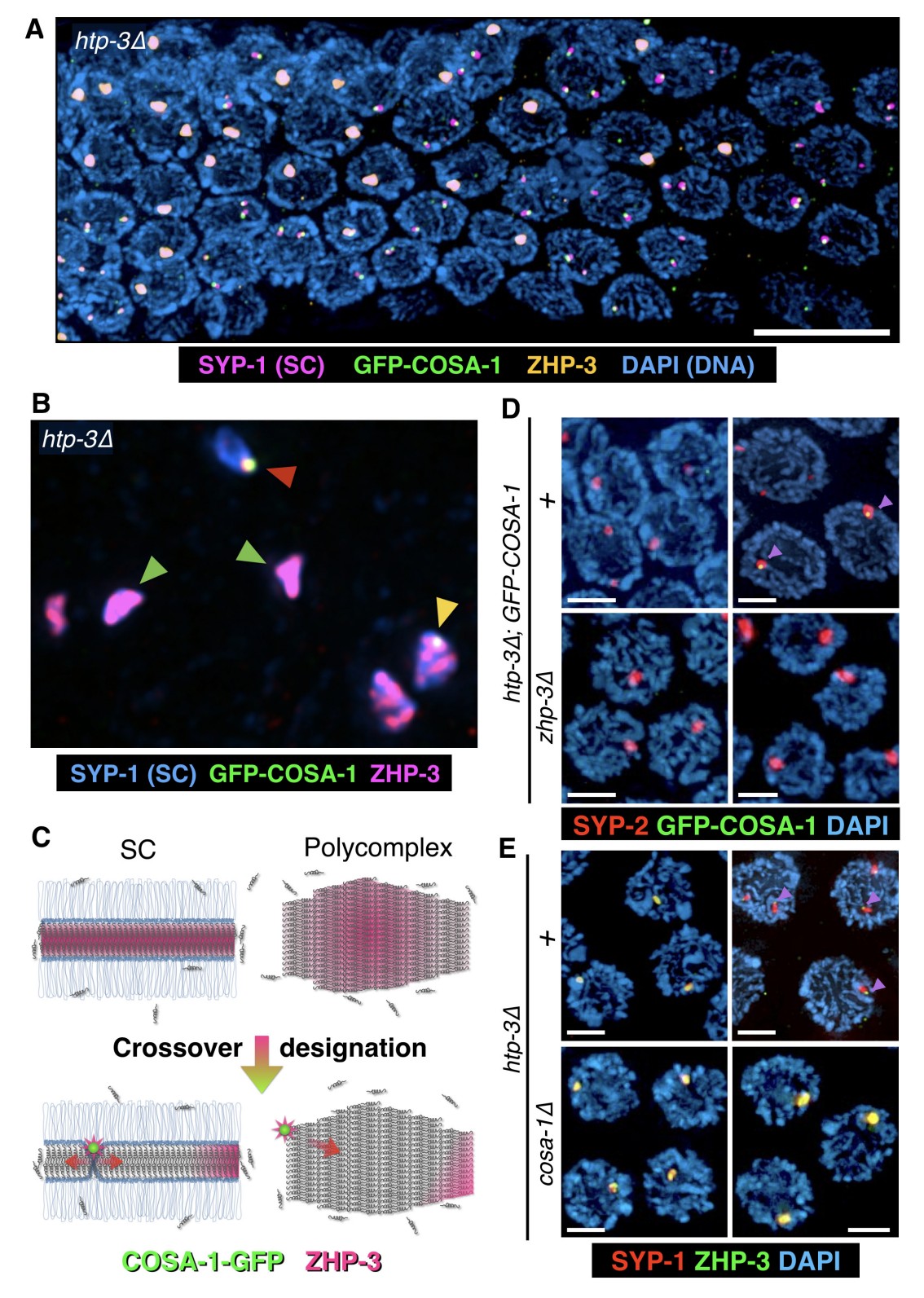

**Figure 6.** The localization of CO factors to polycomplexes recapitulates their dynamic behavior and interdependencies upon CO designation. (A–B) Dynamic relocalization of ZHP-3 and COSA-1 during meiotic progression. Fluorescence micrograph of a representative gonad from *htp-3(tm3655)* hermaphrodite expressing GFP-COSA-1 stained with antibodies against SYP-1 (SC/polycomplexes; magenta), ZHP-3 (orange), and GFP (COSA-1; green), and counterstained with DAPI (blue). ZHP-3 is detected throughout the volume of polycomplexes in the mid-pachytene region of the gonad,

*Figure 6 continued on next page*

*Figure 6 continued*

while GFP-COSA-1 is not detected. In nuclei at the 'loop' region of the gonad, where nuclei normally exit pachytene (see *Figure 2B* for reference), ZHP-3 staining becomes confined to small foci abutting polycomplexes, where it colocalizes with GFP-COSA-1. An intermediate stage in which COSA-1 and ZHP-3 colocalize at the surface but ZHP-3 also remains throughout the polycomplex is also seen in some nuclei. Scale bar = 10 μm. (B) A higher magnification image of a similarly stained gonad, except SYP-1 is shown in blue and ZHP-3 is shown in magenta (DAPI is not shown). Early, late and intermediate stage polycomplexes are marked with green, red and yellow arrowheads, respectively. (C) A model of ZHP-3 and COSA-1 dynamic localization to SCs and polycomplexes. See main text for details. (D) Early (left) and late (right) stage pachytene nuclei from adult *htp-3* or *htp-3 zhp-3* hermaphrodites expressing GFP-COSA-1. Gonads were dissected and stained with antibodies against SYP-2 (SC/polycomplexes; red) and GFP (COSA-1; green) and counterstained with DAPI (blue). GFP-COSA-1 does not form foci associated with polycomplexes in the absence of ZHP-3. Scale bars = 2 μm. (E) Early (left) and late (right) stage pachytene nuclei from adult *htp-3* or *htp-3; cosa-1* hermaphrodites. Gonads were dissected and stained with antibodies against SYP-1 (SC/polycomplexes; red) and ZHP-3 (green), and counterstained with DAPI (blue). In the absence of COSA-1, ZHP-3 remains localized throughout polycomplexes even at late stage of meiotic prophase. Scale bars = 2 μm. The entire gonads for (D) and (E) are shown in *Figure 6—figure supplement 1C–D*.

The following figure supplement is available for figure 6:

**Figure supplement 1.** Further characterization of the localization and 1,6-hexanediol sensitivity of ZHP-3 and COSA-1.

implicated in regulating late meiotic prophase events in *C. elegans* (*Nadarajan et al., 2016*), does not affect the relocalization of COSA-1 or ZHP-3 on SCs or polycomplexes (Liangyu Zhang, OR and AFD, unpublished), so the nature of the 'exit pachytene' signal that triggers this switch is unknown.

Finally, we note that long-range order within liquid crystals allows propagation of structural transitions, known as 'disclinations,' over long distances (*Rey, 2010*; *Bukusoglu et al., 2016*), similar to a 'domino effect.' Energy can also be stored within liquid crystals as elastic strain, and released through abrupt topological changes (*Bukusoglu et al., 2016*). This property could provide a rapid response to crossover formation. Specifically, we imagine that structural changes could be induced within the SC at sites of CO formation and transmitted laterally, in which case interference might be mediated by a mechanochemical rather than a purely biochemical signal. This idea is analogous to a 'beam-film' model that has been proposed to explain interference (*Zhang et al., 2014*), but posits that the medium that stores strain is the SC, or more precisely, the SC as confined by its interaction with chromosome axes. Ultrastructural analysis of SC organization before and after CO formation may eventually allow this idea to be tested. Additionally, future studies of the role of SC organization and dynamics in CO interference may enable development of new synthetic materials with long-range signaling capabilities.

## Materials and methods

### Worm strains and transgenes

All strains were cultured using standard methods (*Brenner, 1974*). Worms were maintained at 20°C, except that animals expressing *mMaple-HIM-3* and *mMaple-SYP-3* were cultured at 25°C to enhance transgene expression. Where indicated, worm plates were incubated overnight at 26.5°C to induce SC aggregation (*Bilgir et al., 2013*). See *Table 1* for a full list of the strains used.

*HTP-3-mRuby* (*ieSi17*) and *mRuby-SYP-3* (*ieSi19*) were constructed and inserted using MosSCI, essentially as previously described for GFP-SYP-3 and HTP-3-GFP (*Kim et al., 2014*; *Rog and Dernburg, 2015*). A transgene including a 3x(Gly-Gly-Ser-Gly) linker and mRuby (*Kredel et al., 2009*) immediately downstream of the coding sequnce of *htp-3* was constructed using the MosSCI repair template pCFJ178. This construct was integrated at the *cxTi10882 IV* site by MosSCI (*Frøkjaer-Jensen et al., 2008, 2012*), and later homozygosed. *mMaple3-SYP-3* was constructed using a codon-optimized synthetic DNA sequence, with modifications to improve germline expression (*Wang et al., 2014b*; *Frøkjær-Jensen et al., 2016*), and inserted by MosSCI. All SYP-3 and HTP-3 transgenes were crossed into mutants lacking the corresponding wild-type proteins (*syp-3(ok758)* or *htp-3(tm3655)*, respectively). *mMaple3-HIM-3* was constructed by inserting a codon-optimized *mMaple3* sequence at the N-terminus of the endogenous *him-3* gene using CRISPR-Cas9 (*Dickinson et al., 2013*). Analysis of *htp-3* mutant animals was performed on homozygous progeny of balanced heterozygotes, and their heterozygous siblings served as wild-type controls. A *cosa-1*

**Table 1.** Strains used in this study.

| Strain Name | Genotype |
| --- | --- |
| CA257 | him-8(tm611) IV |
| CA277 | unc-24(e138) him-8(e1489) spo-11(ok79) / mIs11 IV |
| CA795 | htp-1(gk174) / nT1 [unc-?(n754) let-?] (IV;V) |
| CA821 | htp-3(tm3655) I / hT2 [bli-4(e937) let-?(q782) qIs48] (I;III) |
| CA826 | cra-1(tm2144) III / hT2 [bli-4(e937) let-?(q782) qIs48] (I;III) |
| CA861 | syp-3(ok758) I / hT2 [bli-4(e937) let-?(q782) qIs48] (I;III) |
| CA899 | syp-3(me42) I / hT2 [bli-4(e937) let-?(q782) qIs48] (I;III) |
| CA904 | rec-8(ok978) / nT1 IV; coh-4(tm1857) coh-3(gk112) / nT1 [qIs51] V |
| CA1010 | meIs8 [pie-1p::GFP::cosa-1 + unc-119(+)] II |
| CA1095 | rec-8(ok978) IV / nT1[qIs51] (IV;V) |
| CA1122 | cosa-1(me13) / qC1 [dpy-19(e1259) glp-1(q339) qIs26] III |
| CA1234 | syp-3(ok758) I; ieSi63 [cbunc-119+, psyp-3::mMaple3::syp-3] II; unc-119(ed3) III. |
| CA1237 | htp-3(tm3655) I; cosa-1(ie98) III |
| CA1238 | htp-3(tm3655) I/hT2 [bli-4(e937) let-?(q782) qIs48] (I,III); meIs8 [pie-1p::GFP::cosa-1 + unc-119(+)] II |
| CA1239 | htp-3(tm3655) zhp-3(ie97) I; meIs8 [pie-1p::GFP::cosa-1 + unc-119(+)] II |
| CA1253 | syp-3 (ok758) I; ieSi11 [cbunc-119+, Psyp-3::EmeraldGFP::syp-3] II; unc-119(ed3) III |
| CA1255 | htp-3(tm3655) syp-3(ok758) I; ieSi11 [cbunc-119+, Psyp-3::EmeraldGFP::syp-3] II; ieSi17 [cbunc-119, Phtp-3::htp-3::mRuby] IV |
| CA1255 | htp-3(tm3655) syp-3(ok758) I; ieSi11 [cbunc-119+, Psyp-3::EmeraldGFP::syp-3] II; ieSi17 [cbunc-119, Phtp-3::htp-3::mRuby] IV |
| CA1297 | syp-3(ok857) I; ieSi11 [cbunc-119+, Psyp-3::EmeraldGFP::syp-3] II; him-8(tm611) IV |
| CA1298 | meIs9 [gfp::syp-3]; unc-119 |
| CA1299 | hal-2(me79) / qC1 [dpy-19(e1259) glp-1(q339) qIs26] III; meIs9 (gfp::syp-3); unc-119 |
| CA1300 | syp-3(ok857) htp-3(tm2655) I; ieSi11 [cbunc-119+, Psyp-3::EmeraldGFP::syp-3] II |
| CA1303 | syp-3 (ok857) I; ieSi11 [cbunc-119+, Psyp-3::EmeraldGFP::syp-3] II; htp-1(gk174)/nT1 [unc-?(n754) let-?] (IV;V) |
| CA1309 | unc-119(ed3) III; ieSi21[cbunc-119+, Psun-1::sun-1::mRuby] IV |
| CA1350 | him-3(ie34[mMaple3::him-3]) IV |

(ie98) htp-3(tm3655) strain was constructed by inserting a premature stop codon to disrupt the cosa-1 gene using CRISPR-Cas9 in htp-3(tm3655)/hT2 animals. A zhp-3(ie97) htp-3(tm3655); COSA-1-GFP strain was constructed by inserting a premature stop codon disrupting the zhp-3 gene using CRISPR-Cas9 in COSA-1-GFP; htp-3(tm3655)/hT2 animals.

## Time-lapse imaging

Adult hermaphrodites were immobilized for imaging as previously described (*Rog and Dernburg, 2015*). Briefly, worms were placed on freshly prepared agarose pads (7.5% in water) overlaid with 100 nm polystyrene beads (Polysciences, cat#00876) (*Kim et al., 2013*). The pad and beads included freshly-dissolved serotonin creatinine sulfate (Sigma-Aldrich, St. Louis, MO) at a final concentration of 25 mM. Worms were overlaid with high-performance coverslips (0.17 ± 0.005 mm; Schott) that were sealed to the slide with VALAP (1:1:1 vaseline:lanolin:paraffin), and imaged immediately afterwards. Time-lapse images were recorded from the posterior gonad arm, the motion of which was less perturbed by pharyngeal pumping than the anterior arm. Only gonads for which chromosome motion was detected during the entire recording were analyzed, except where otherwise indicated.

Hexanediol treatment during time-lapse imaging was performed by dissecting adult hermaphrodites in 30 μL 1X egg buffer (EB) on poly-L-lysine covered coverslips (Neuvitro, Vancouver, WA). Extruded gonads were gently pressed against the coverslip to promote adhesion. At the indicated times, an equal volume of hexanediol in EB was added to the drop to attain the indicated final

concentrations of hexanediol. Where indicated, the buffer was subsequently diluted with additional EB lacking hexanediol.

Images were acquired using a Marianas spinning-disc confocal microscope (Intelligent Imaging Innovations, Inc. [3i], Denver, CO) at ambient temperature (19°C–24°C), using a 100 × 1.46 NA oil immersion objective, yielding a pixel spacing of 133 × 133 nm. 3D image stacks of 11–17 sections at 0.5 μm z-spacing were acquired over 1–3 s. Raw image stacks were analyzed using Imaris 7.3 or 8.0 (Bitplane AG, Zurich, Switzerland). Time-lapse series were segmented, tracked and aligned based on overall fluorescence using the Spots tool. Compactness was measured using the Surface tool, analyzing at least three different gonads for each condition.

## Protein mobility measurements

Photoconversion of mMaple3 fusion proteins was performed using the Marianas spinning-disc confocal system described above, equipped with a Vector FRAP module. Two-color 3D image stacks of 11 sections at 0.8 μm z-spacing were acquired over 1–3 s. The first time point included activation of 10 × 10 pixel regions (1.33 × 1.33 μm) using a 405 nm laser at each focal position. Following photoconversion, 3D stacks were acquired every 1–5 min, as indicated.

Our recordings indicate that photoconverted molecules can spread both along SCs and between SCs on different chromosomes. We thus assume that this movement has multiple rate components. At the temporal and spatial resolution we could achieve in living animals with spinning disk confocal microscopy, it was not possible to directly measure diffusion rates or the surface tension of individual SC compartments. Instead, we compared the spreading rates of SYP-3 (SC) and HIM-3 (axis) proteins by calculating the rate of change of the volume of the nucleus containing photoconverted (red) signal, based on 3D images from time-lapse recordings. The volume of the nucleus occupied by photoconverted molecules at each time point was placed in one of four bins (25%, 50%, 75% or 100%), which were then plotted as a function of time elapsed since photoconversion. At least 15 nuclei were scored for each fluorescent marker. The expansion rate was defined as the slope of a line fitted to the data. These values were compared to simple simulations in which a point source of fluorescence became homogeneously distributed throughout the nucleus in 1 hr.

## Immunofluorescence

Immunofluorescence was performed as previously described (*Phillips et al., 2009*), except that where indicated, dissected hermaphrodites were briefly incubated with 1,6-hexanediol in egg buffer immediately before adding fixative. The following antibodies, all of which have been previously described, were used: anti-SYP-2 (rabbit, affinity purified, 1:1,000), anti-SYP-1 (goat, affinity purified, 1:1,000), anti-GFP (mouse monoclonal, Roche, Indianapolis, IN, 1:400), anti-SUN-1 (rabbit, affinity-purified, Novus Biologicals, Littleton, CO, 1:10,000), anti-HTP-3 (guinea pig, 1:500), anti-HIM-8 (rat, 1:500), anti-ZHP-3 (rabbit, affinity-purified, Novus Biologicals, 1:10,000) and secondary antibodies conjugated to Alexa 488, Cy3, or Cy5 (Jackson ImmunoResearch, West Grove, PA or Molecular Probes, Thermo-Fisher, Waltham, MA; 1:300). Imaging was performed with a Marianas spinning-disc confocal microscope system (Intelligent Imaging Innovations, Inc. [3i]) using a 100 × 1.46 NA oil objective, or with a DeltaVision Elite system (GE Healthcare, Pittsburgh, PA) equipped with an Olympus 100 × 1.4 NA oil-immersion objective. Wide-field images were deconvolved using the SoftWoRx Suite (GE Healthcare). 3D stacks were visualized in Imaris 7.3 or 8.0 (Bitplane).

To evaluate the sensitivity of the SC to hexanediol in various mutants (*Figure 4—figure supplement 2A*), at least 3 slides with >10 gonads per slide were scored. The data are presented as the average value for a set of slides. Gonads were scored as 'dissolved' if no filaments were visible, other than short SC stretches on the X chromosome and/or within apoptotic nuclei.

## Hexanediol treatment and imaging of meiotic yeast cells and *Drosophila* oocytes

The following yeast strains were used:

**Brün292**: *MATa/alpha; ndt80::LEU2:ndt80::LEU2 flo8::KanMX6/flo8::KanMX6 ZIP1::GFP(700)/ ZIP1::GFP(700)* (SK1 strain background)

**Brün793**: *MATa/alpha; GAL-NDT80::TRP1/GAL-NDT80::TRP1 ura3::pGPD1-GAL4(848).ER:: URA3/ura3::pGPD1-GAL4(848).ER::URA3 leu2::URA3p-tetR-tdTomato::LEU2 CENV::tetOx224::HIS3 REC8-GFP-URA3/REC8-GFP-URA3* (SK1 strain background)

Sporulation and pachytene arrest of *ndt80* strains was was carried out as described (*Carlile and Amon, 2008*). Where indicated, 1,6-hexanediol diluted in sporulation medium was added to cells immediately prior to imaging. Where indicated, cells were treated with various concentration of 1,6-hexanediol and KCl by mixing a 2x concentrated solution with cells washed 3 times with 20 mM phosphate buffer (pH 7). 'Washing' of hexanediol was performed by mixing the treated cells with 10x excess of 20 mM phosphate buffer (pH 7). Unfixed cells were imaged using a Marianas spinning-disc confocal microscope, as described above.

To analyze the effects of hexanediol on *Drosophila melanogaster* SCs, ovaries from young, mated *c(3)G-GFP; c(3)G/+* females fed on yeast paste were dissected in PBS (*Page and Hawley, 2001*), separated into ovarioles, and transferred to a drop of PBS in a glass-bottomed petri dish. 3D wide-field images were acquired using a DeltaVision microscope equipped with a 100 × 1.4 NA objective. An equal volume of 2x 1,6 hexanediol in PBS was added during imaging. For immunofluorescence, ovarioles from *c(3)G-GFP; c(3)G/+* or from *GAL4-nos/+; UASp-c(2)M-3xHA/+* females were dissected in Modified Robb's Saline (MRS), incubated briefly with hexanediol in MRS (or just MRS) and fixed with 4% formaldehyde in MRS, following by immunostaining as described (*McKim et al., 2009*). The following antibodies were used: anti-GFP (1:400, mouse, Roche), anti-CID (1:500, chicken), anti-C(2)M (1:500, rabbit), anti-HA (mouse, 1:500), anti-C(3)G (Rabbit, 1:3000), and appropriate secondary antibodies conjugated to dyes (1:300). Samples were imaged on a Marianas spinning-disc confocal microscope, as described above.

## Electron microscopy

*C.elegans* adult hermaphrodites were high-pressure frozen and freeze-substituted according to previously described methods (*McDonald and Webb, 2011*; *McDonald, 2014*). Briefly, worms were frozen in 50 μm deep specimen carriers in a HPM-010 high pressure freezer (Bal-Tec, Balzers, Liechstenstein), freeze-substituted in 1% osmium tetroxide plus 0.1% uranyl acetate in acetone over 2.5 hr, and embedded in Epon-Araldite resin over a period of 2.5 hr. Serial sections (70 nm thick) were picked up on slot grids, post-stained with uranyl acetate and lead citrate, and imaged in Tecnai 12 transmission electron microscope operating at 120kV (FEI, Hillsboro, OR), using an Ultrascan 1000 charge coupled device camera (Gatan, Pleasanton, CA).

## Movie legends

All movies show maximum-intensity projection images from 3D time-lapse recordings of *C. elegans* gonads acquired with a spinning disk confocal microscope. In most cases the worms were intact and alive during recording; in experiments involving 1,6-hexanediol treatment, gonads were extruded by nicking the cuticle with a scalpel blade shortly before imaging began. Elapsed times are indicated at the bottom right corner of each frame as hours:minutes:seconds:(milliseconds).

## Acknowledgements

We thank Kent McDonald for assistance with high-pressure freezing and electron microscopy, Christian Frøkjær-Jensen for providing unpublished reagents, Gloria Brar, Nicole Beier, Anne Villeneuve, Kim McKim, Mary Lilly, and Keith Cheveralls for technical advice, assistance, and reagents, and Yuval Mazor, Rohit Pappu, Alexandra Zidovska, and members of our laboratory for helpful discussions and critical reading of the manuscript. Some nematode strains used in this work were provided by the Caenorhabditis Genetics Center, which is funded by the NIH - Office of Research Infrastructure Programs (P40 OD010440). This work was supported by a European Molecular Biology Organization Long-Term Fellowship (ALTF 564–2010) to OR, and support to AFD from the National Institutes of Health (R01 GM065591) and the Howard Hughes Medical Institute.

## Additional information

### Funding

| Funder | Grant reference number | Author |
| --- | --- | --- |
| European Molecular Biology Organization | ALTF 564-2010 | Ofer Rog |
| Howard Hughes Medical Institute | | Abby F Dernburg |
| National Institutes of Health | GM065591 | Abby F Dernburg |

The funders had no role in study design, data collection and interpretation, or the decision to submit the work for publication.

### Author contributions

OR, Conceptualization, Resources, Data curation, Software, Formal analysis, Validation, Investigation, Visualization, Methodology, Writing—original draft, Writing—review and editing; SK, Resources, Investigation, Methodology; AFD, Conceptualization, Resources, Data curation, Software, Formal analysis, Supervision, Funding acquisition, Validation, Investigation, Visualization, Methodology, Writing—original draft, Project administration, Writing—review and editing

### Author ORCIDs

Abby F Dernburg, http://orcid.org/0000-0001-8037-1079

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
