## [Decision Letter]

Thank you for submitting your article "The synaptonemal complex is a liquid crystalline compartment that spatially regulates meiotic recombination factors" for consideration by *eLife*. Your article has been favorably evaluated by Anna Akhmanova as the Senior Editor and three reviewers, including Michael K Rosen (Reviewer #3) and a member of our Board of Reviewing Editors.

The reviewers have discussed the reviews with one another and the Reviewing Editor has drafted this decision to help you prepare a revised submission.

In the past few years, liquid-liquid de-mixing or phase separation of proteins has emerged as a novel way to think about concentrating biomolecules in a heavily crowded cytoplasmic microenvironment. In this manuscript by Rog et al., the authors have shown the synaptonemal complex (SC), that for years was thought to be a solid polymeric complex of proteins, because of the striated structures observed by EM, in fact possess properties reminiscent of a liquid crystal.

PCs can fuse with each other, distort, and condense from diffuse molecules into dense puncta. Similarly, the SC protein SYP-3 (but not the axis protein HIM-3) spreads rapidly in the interchromosomal space, and SCs form through a process consistent with condensation. The authors show that both SCs and polycomplexes are reversibly sensitive to hydrophobic alcohols in a manner similar to some other phase separated cellular structures.

The idea that the SC complex forms a liquid-crystal like structure is provocative, and as you say in the paper, helps to explain some of the biology of these complexes. The reviewers engaged in extensive discussion about the use of the term liquid-crystal.

On the one hand, it certainly helps to unify the observations seen by electron microscopy, with the dynamic observations that you have made in your paper.

On the other hand, there was some discomfort with the strength of the claim that it *is* a liquid crystal, i.e. the same underlying organizational principles as in nonliving smectic LCs. We would therefore recommend softening the language throughout the manuscript, including further discussion of what may be different between these structures and classical LCs.

For instance, in comparison with other intracellular phase separated liquids recently described, which typically exhibit molecular turnover (e.g. from FRAP) on timescales of tens of seconds, SC only exhibits turnover on timescales of tens of minutes. If it is truly a liquid crystal, then one might expect faster dynamics at least within the liquid phase plane. However, while the recovery rate is slower than seen with 3D structures, it is hard to know what one should expect for a very thin strand of liquid flowing in a minute space between two chromosomes; viscosity could be very high. Certainly, a solid would not show spreading at all. The structural similarity and dynamics make, in my mind, a pretty strong analogy between the polycomplexes and SCs. These I think are points worth making in the Discussion.

Another point of discussion was that liquid crystals are generally multilayer. If I understand correctly, the beautiful EM images of polycomplex bodies certainly have such a multilaminar structure, but the normal SC does not (so I would actually not call a typical biological membrane a smectic liquid crystal as stated in the discussion, although multilaminar bilayers can be formed which can be referred to as liquid crystalline arrays. The EM data in Figure 1 show that the polycomplexes are composed of ordered bands with 97 nm spacing, each of which is composed of a striated array. The dynamics show that they fuse with each other, and recover from hexanediol like a liquid. This would be consistent with a smectic LC. Your EM of the SC in the top image of Figure 1 looks very much like a single band taken from the polycomplex – it is also 97 nm thick, and is composed of an analogous striated array. The spacing between the striations looks similar, although that's harder to tell. Again, I think you should comment on these issues, as they will certainly be issue that readers of the paper would question.

While many of the behaviors of SC and polycomplexes are consistent with liquid phase separation, the following description: "In nuclei that had initiated synapsis before arresting, we observed that diffuse GFP-SYP-3 gradually coalesced to join existing segments of SC between chromosomes. While these pre-existing SC segments became much brighter following arrest, they did not spread longitudinally along partially synapsed chromosomes." Once phase separation has occurred, unless the properties of the molecules change (e.g. by phosphorylation), the concentration in the new compartment will not change as new molecules are added, the compartment will merely grow larger. This may not appear to be the case, however, if the initial droplets are smaller than the diffraction limit (as would seem to be the case in Figure 2—figure supplement 1, circle E). In such a case their sub-diffraction growth will appear to be an increase in concentration. Could this be the case here, with multiple short SC segments growing within diffraction-limited volumes?

We would also encourage you to comment on the relationship to surfactants. Given the recent work from the Gerlich group on the chromosome surfactant-like molecule ki-67, this seems important.

Importantly, I think that the title as shown is too strong. To say it "is" a liquid crystal at this point would I think bring unnecessary criticism of your nice work. One possibility would be to change "is" to "as".

Or,

"The synaptonemal complex has similarities to a liquid crystalline compartment that spatially regulates meiotic recombination factors"

---

## [Author Response]

*[…] The idea that the SC complex forms a liquid-crystal like structure is provocative, and as you say in the paper, helps to explain some of the biology of these complexes. The reviewers engaged in extensive discussion about the use of the term liquid-crystal.*

*On the one hand, it certainly helps to unify the observations seen by electron microscopy, with the dynamic observations that you have made in your paper.*

*On the other hand, there was some discomfort with the strength of the claim that it is a liquid crystal, i.e. the same underlying organizational principles as in nonliving smectic LCs. We would therefore recommend softening the language throughout the manuscript, including further discussion of what may be different between these structures and classical LCs.*

We’re not entirely sure what the objection is to calling the SC a liquid crystalline compartment – as we understand it, our work establishes that this material has the 2 defining properties of a liquid crystal, i.e., it is comprised of mobile but spatially ordered molecules. When we’ve discussed our findings with polymer chemists and physicists, they have quickly concluded that the SC is a liquid crystal even when we refrain from using this language. We agree that it is probably distinct from liquid crystals as studied under equilibrium conditions, in that it is likely an active material. We are willing to face potential criticism for using this terminology. However, since the Editor and reviewers feel strongly about this issue, we have agreed to change the title to “The synaptonemal complex has liquid crystalline properties and spatially regulates meiotic recombination factors,” almost the same as the second suggestion below. We have also tempered the language somewhat throughout the text.

*For instance, in comparison with other intracellular phase separated liquids recently described, which typically exhibit molecular turnover (e.g. from FRAP) on timescales of tens of seconds, SC only exhibits turnover on timescales of tens of minutes. If it is truly a liquid crystal, then one might expect faster dynamics at least within the liquid phase plane. However, while the recovery rate is slower than seen with 3D structures, it is hard to know what one should expect for a very thin strand of liquid flowing in a minute space between two chromosomes; viscosity could be very high. Certainly, a solid would not show spreading at all. The structural similarity and dynamics make, in my mind, a pretty strong analogy between the polycomplexes and SCs. These I think are points worth making in the Discussion.*

We have now addressed this more directly and extensively in the text, but it is a very complex issue. It’s not clear to what extent molecular diffusion within the synaptonemal complex is analogous to that within amorphous liquids. For example, as alluded to by the reviewer(s), it is not yet clear if diffusion within the SC should be modeled as a 1D or 2D system, or how its confinement between chromosomes might impact mobility. In addition, we cannot currently distinguish between diffusion within an individual SC compartment and exchange of subunits between compartments. We are currently exploring these questions using correlation-based image analysis methods, but they are clearly beyond the scope of this work.

*Another point of discussion was that liquid crystals are generally multilayer. If I understand correctly, the beautiful EM images of polycomplex bodies certainly have such a multilaminar structure, but the normal SC does not (so I would actually not call a typical biological membrane a smectic liquid crystal as stated in the discussion, although multilaminar bilayers can be formed which can be referred to as liquid crystalline arrays. The EM data in Figure 1 show that the polycomplexes are composed of ordered bands with 97 nm spacing, each of which is composed of a striated array. The dynamics show that they fuse with each other, and recover from hexanediol like a liquid. This would be consistent with a smectic LC. Your EM of the SC in the top image of Figure 1 looks very much like a single band taken from the polycomplex – it is also 97 nm thick, and is composed of an analogous striated array. The spacing between the striations looks similar, although that's harder to tell. Again, I think you should comment on these issues, as they will certainly be issue that readers of the paper would question.*

We have tried to address this, and also tried to clarify that polycomplexes have been specifically defined, based on EM images from many organisms (first by Roth, 1966), as aggregates of SC proteins with a periodic internal structure that appear as repeating units of SCs. This is well-known in the meiosis field and unlikely to be questioned by readers familiar with the SC. Ours is the first demonstration that the aggregates that form in worm axis mutants have the same periodic internal structure seen for polycomplexes in other organisms, and that this order is lost at high temperatures. Our EM images obtained from high-pressure-frozen samples are actually somewhat less clear than some of the “classic” images from the mid-20th century using conventional fixation, in which the SC and polycomplexes appear in higher contrast, probably because of more extensive extraction of other cellular material.

*While many of the behaviors of SC and polycomplexes are consistent with liquid phase separation, the following description: "In nuclei that had initiated synapsis before arresting, we observed that diffuse GFP-SYP-3 gradually coalesced to join existing segments of SC between chromosomes. While these pre-existing SC segments became much brighter following arrest, they did not spread longitudinally along partially synapsed chromosomes." Once phase separation has occurred, unless the properties of the molecules change (e.g. by phosphorylation), the concentration in the new compartment will not change as new molecules are added, the compartment will merely grow larger. This may not appear to be the case, however, if the initial droplets are smaller than the diffraction limit (as would seem to be the case in Figure 2—figure supplement 1, circle E). In such a case their sub-diffraction growth will appear to be an increase in concentration. Could this be the case here, with multiple short SC segments growing within diffraction-limited volumes?*

Thanks for pointing out this concern, which reflects a lack of clarity on our part. We didn’t mean to suggest that the segments were becoming denser or more concentrated; we think that they are probably growing in thickness (above and below the plane defined by the SC) but not longitudinally; i.e., along the chromosome axis. We have added some text to explain and justify this idea.

*We would also encourage you to comment on the relationship to surfactants. Given the recent work from the Gerlich group on the chromosome surfactant-like molecule ki-67, this seems important.*

We don’t see an obvious relationship between our findings and the intriguing work on Ki-67. While it’s certainly possible that assembly of the SC helps to overcome a repulsive force between homologous chromosomes arising from “stiff brush”-like behavior, we don’t have any evidence that addresses this, so we have chosen not to address this in the revised manuscript.

*Importantly, I think that the title as shown is too strong. To say it "is" a liquid crystal at this point would I think bring unnecessary criticism of your nice work. One possibility would be to change "is" to "as".*

*Or,*

*"The synaptonemal complex has similarities to a liquid crystalline compartment that spatially regulates meiotic recombination factors"*

See our comments above about this.